# Estimation of turbulent heat flux over leads using satellite thermal images

Meng Qu[1], Xiaoping Pang[1], Xi Zhao[1, 2], Jinlun Zhang[2], Qing Ji[1], Pei Fan[1]

[1]Chinese Antarctic Center of Surveying and Mapping, Wuhan University, Wuhan, China

[2]Applied Physics Laboratory, Polar Science Center, University of Washington, Seattle, WA, USA

*Correspondence to*: Xi Zhao (xi.zhao@whu.edu.cn)

**Abstract.** Sea ice leads are an important feature in pack ice in the Arctic. Even covered by thin ice, leads can still serve as prime windows for heat exchange between the atmosphere and the ocean, especially in the winter. Lead geometry and distribution in the Arctic have been studied using optical and microwave remote sensing data, but turbulent heat flux over leads has only been measured onsite during a few special expeditions. In this study, we derive turbulent heat flux through leads at different scales using a combination of surface temperature and lead distribution from remote sensing images and meteorological parameters from a reanalysis dataset. First, ice surface temperature (IST) was calculated from Landsat-8 Thermal Infrared Sensor (TIRS) and Moderate Resolution Imaging Spectroradiometer (MODIS) thermal images using a split-window algorithm; then, lead pixels were segmented from colder ice. Heat flux over leads was estimated using two empirical models: bulk aerodynamic formulae and a fetch-limited model with lead width from Landsat-8. Results show that even though the lead area from MODIS is a little larger, the length of leads is underestimated by 72.9% in MODIS data compared to TIRS data due to the inability to resolve small leads. Heat flux estimated from Landsat-8 TIRS data using bulk formulae is 56.70% larger than that from MODIS data. When the fetch-limited model was applied, turbulent heat flux calculated from TIRS data is 32.34% higher than that from bulk formulae. In both cases, small leads accounted for more than a quarter of total heat flux over leads, mainly due to the large area, though the heat flux estimated using the fetch-limited model is 41.39% larger. A greater contribution from small leads can be expected with larger air–ocean temperature difference and stronger winds.

## 1 Introduction

Leads are linear structures of the ocean surface within pack ice that are exposed to the atmosphere during an opening event caused by various forces, such as wind and water stresses. In winter, thin ice forms quickly in newly opened leads due to the large temperature difference between the ocean and the atmosphere (Kwok, 2001). The opening of leads breaks the continuity of insulating ice and creates windows for a stronger air–ocean interaction. Newly opened leads are the main source of ice production, brine rejection, and heat transfer from the ocean to the atmosphere (Alam and Curry, 1998). Turbulent heat flux over open water could be two orders of magnitude larger than that through mature ice (Maykut, 1978). Although decreasing rapidly with growing ice thickness, ice growth rates can still be an order of magnitude larger for 50 cm thick young ice than for 3 m thick ice (Maykut, 1986). In the central Arctic, open water usually comprises no more than 1% of the ice pack area during the winter. However, open water, together with thin ice (<1 m) estimated to be 10% of the whole ice area, contributes to more than 70% of the upward heat flux (Maykut, 1978; Marcq and Weiss, 2012). A model study shows that an increased lead fraction by 1% can lead to local air temperature warming up to 3.5 K in winter (Lüpkes et al., 2008).

Leads also allow more surface absorption of radiation due to their lower albedo compared to thick ice. This will accelerate sea ice thinning in summer and delay refreezing in early winter, therefore decrease the mechanical strength of the ice cover and allow even more fracturing, larger drifting speed and deformation, and faster export of sea ice to lower latitudes (Rampal et al., 2009). As the ice pack gets thinner (Kwok and Rothrock, 2009) and more mobile (Spreen et al., 2011), favorable for deformation and opening, networks of more intensive lead with stronger local influence are expected.

Since the late 1970s, remote sensing images obtained by satellite sensors, including optical, thermal, and microwave, have been used to detect sea ice leads in the Arctic (Fetterer and Holyer, 1989; Fily and Rothrock 1990; Fett et al., 1997). Lindsay

and Rothrock (1995) promoted the concept of potential open water for lead detection, which requires both temperature and albedo differences between ice surface pixels and open water tie-points. Based on different emissivities of thin ice at two microwave frequencies available for the Advanced Microwave Scanning Radiometer for the Earth Observing System (AMSR-E), Röhrs and Kaleschke (2012) developed a retrieval algorithm to estimate Arctic lead concentration, similar to sea ice concentration. The algorithm could provide subpixel information on lead distribution, but the resolution is still too coarse to detect small leads prevailing in pack ice. Willmes and Heinamann (2015) mapped pan-Arctic lead distribution at 1 km resolution using local temperature anomaly $\Delta T_s$ to identify leads from surrounding thick ice. Other remote sensing data, including altimetry, high-resolution optical, and synthetic-aperture radar (SAR) images, were also used to identify leads in limited areas due to constraints of cloud contamination and data acquisition restrictions (Key et all., 1993; Miles and Barry, 1998; Kwok, 2001; Weiss and Marsan, 2004; Wernecke and Kaleschke, 2015; Murashkin et al., 2018).

Regardless of spectral characteristics used for lead detection, the scale dependence of lead statistics was explored in a few studies (Key et al., 1994; Weiss and Marsan, 2004; Marson et al., 2004). Key et al. (1994) studied the effects of the sensor's field of view (FOV) using degraded optical images from the Landsat Multi-Spectral Scanner (MSS). They suggested that the mean lead width expands, and the lead fraction drops as the pixel size builds up in gradually degraded images. Assuming higher heat flux over narrow leads than wider leads, estimated turbulent heat flux was reduced by 45% as the FOV was degraded from 80 m to 640 m, mainly due to reduced lead fraction.

Bulk aerodynamic formulae are frequently used in climate models to generalize the turbulent heat flux from open water in Arctic pack ice (Lindsay and Rothrock, 1994; Walter et al. 1995). The bulk formulae attribute heat flux over leads to wind speed, temperature differences between the surface and the atmosphere, and a turbulent transfer coefficient for heat, which is a function of the stability of the near-surface atmosphere and the roughness of the surface. In this approach, Lindsay and Rothrock (1994) estimated sensible heat flux using surface temperature retrieved from the Advanced Very-High-Resolution Radiometer (AVHRR). While observations suggest that for small leads, down to dozens of meters in width, the discontinuity between leads and pack ice causes the creation of a thermal internal boundary layer (TIBL) in the bottom atmosphere, reducing turbulent heat exchange on the downwind side (Venkatram, 1977; Andreas et al., 1979). Convective plumes formed above leads may further complicate the process within the TIBL (Tetzlaff et al., 2015).

Models were developed for estimation of TIBL thickness and turbulent heat flux over coastal polynyas, leads, and ice edges (Alam and Curry, 1997; Andreas and Cash, 1999; Renfrew and King, 2000, Chechin and Lüpkes, 2017). Chechin and Lüpkes (2017) modeled boundary layer development downwind of the ice edge, potential temperature, and mix-layer height, and wind speed variation was analyzed as well. Renfrew and King (2000) modeled turbulent heat flux over large fetch (5–50 km wide, typical for coastal polynya) during cold-air outbreaks. The dependence of turbulent heat flux on lead width was estimated in several studies (Andreas and Murphy, 1986; Alam and Curry, 1997; Andreas and Cash, 1999). On the basis of the Monin–Obukhov similarity theory and the surface renewal theory, Alam and Curry (1997) estimated turbulent heat flux over leads using an intricate surface roughness model (Bourassa et al., 2001). Sensible heat flux across a single lead is integrated from fetch 0 to fetch $X$. Andreas and Murphy (1986) calculated transfer coefficient $C_{N10}$ at 10 m height for turbulent heat in neutral stability, using the nondimensional fetch $-X/L$, where $L$ is the Obukhov length. A maximum $C_{N10}$ of $1.8 \times 10^{-3}$ was found at small fetch, and the value decrease to $1.0 \times 10^{-3}$ with increasing $-X/L$. Andreas and Cash (1999) computed lead-average turbulent heat flux using transfer coefficient $C_*$ as a function of stability parameter $-h/L$, where $h$ is the fetch-dependent height of the TIBL. For small fetch ($-h/L < 6$), turbulent heat is exchanged by mixed free and forced convection, resulting in a large $C_*$ and higher heat flux.

A power law distribution of lead widths was also reported in various studies (Wadhams, 1981; Wadhams et al., 1985; Lindsay and Rothrock, 1995), indicating that small leads prevail in the Arctic. Impacts of lead width on heat flux were reported by Maslanik and Key (1995) and Marcq and Weiss (2012) using different width distribution models. However, fetch-limited models have not been applied to surface temperature fields retrieved from remote sensing imagery to estimate turbulent heat flux at regional scale, due to the coarse resolution of operational thermal sensors. Fortunately, the launch of Landsat-8 in February 2013 has provided a unique opportunity for estimation of turbulent heat flux with finer-resolution temperature fields.

In this paper, we derive lead distribution using thermal images from two sensors, MODIS and TIRS aboard Terra and Landsat-8, respectively, at different resolution scales. Then we estimate heat flux over leads with remote sensing temperature

fields using both the bulk formulae and a fetch-limited model proposed by Andreas and Cash (1999). With the result, we analyze how the scale property of leads may affect the calculation of heat exchange through leads.

## 2 Data

Three successive scenes of Level 1 terrain-corrected (L1T) Landsat-8 TIRS images and one corresponding MODIS image acquired on April 26, 2016, were used in this study (Table 1). As shown in Fig. 1, the mosaic image of the three TIRS scenes covers an area of about 98,000 km² in the marginal ice zone (MIZ) in east Beaufort Sea, with floes and leads of various lengths and widths spread in the region. We obtained corresponding 10 m wind vector, 2 m air temperature, and dew point temperature from the European Center for Medium-Range Weather Forecasts (ECMWF) ERA-interim reanalysis dataset. This dataset provides global coverage with a temporal resolution of 3 hours, 0.125° grid data is available for download (~10 km in study area, interpolated from original 0.75° grid). The time difference between reanalysis data and Landsat-8 or MODIS images is within half an hour.

Willmes and Heinamann (2015) used the MOD29 ice surface temperature (IST) product (Hall and Riggs, 2015) from the National Snow and Ice Data Center (NSIDC) to retrieve leads. The MOD29 product is filtered for cloud contamination using a cloud mask from MOD35. However, inspection of the corresponding MOD29 map of the study area revealed that the pixels within leads marked as clouds are likely open water with ocean fog or plume over the surface (Fett et al., 1997). Apart from that, the study area within the Landsat-8 frame is mostly unobstructed by clouds. To preserve potential lead areas, we applied the NSIDC algorithm (Hall et al. 2001) to thermal images from MODIS L1B to calculate IST instead of using the MOD29. Therefore, no cloud mask procedure was performed in our study.

The Landsat-8 satellite is in the same near-polar, sun-synchronous, 705 km circular orbit and position as the Landsat-5 satellite decommissioned in 2013. Landsat-8 data are acquired in 185 km swaths and segmented into 185 km × 180 km scenes defined in the second Worldwide Reference System (WRS-2) of path (ground track parallel) and row (latitude parallel) coordinates (Arvidson et al., 2001). The TIRS instrument, a major payload aboard Landsat-8, can observe the ocean surface at a resolution of 100 m by using split-window thermal infrared bands, comparable to MODIS thermal infrared bands, at a resolution of 1000 m. The two narrower thermal infrared channels in the atmospheric window enable application of the widely used split-window algorithm (SWA) in IST retrieval rather than the single-channel method used for TIRS predecessors.

Note that in L1T product, the TIRS bands at 100 m resolution were resampled to 30 m by cubic convolution and co-registered with the Operational Land Imager (OLI) spectral bands. Apart from the TIRS thermal bands, the top of atmosphere reflectance from the Landsat-8 near-infrared band was used for classification between ice and open water in surface temperature retrieval. A panchromatic band with a resolution of 15 m was used as validation data for lead detection in this study.

## 3 Method

### 3.1 IST Retrieval

Key et al. (1997) developed an SWA for IST retrieval from AVHRR, and the algorithm was then adapted for MODIS thermal images with a different set of coefficients (Hall et al. 2001). The equation is as follows:

$$\text{IST} = a + bT_{31} + c(T_{31} - T_{32}) + d[(T_{31} - T_{32})(\sec q - 1)] \tag{1}$$

where $T_{31}$ and $T_{32}$ are brightness temperature from MODIS thermal bands B31 and B32; $a$, $b$, $c$, and $d$ are coefficients developed for specific sensors using a radiance transfer model; $q$ represents the incidence angle; and $\sec q$ is the secant of $q$.

Since there is no special SWA available for sea ice surface temperature retrieval from Landsat-8, a land surface temperature formulation (Du et al. 2015) developed for a wide range of surface types, including ice and snow, was used:

$$T_s = b_0 + \left(b_1 + b_2 \frac{1-\varepsilon}{\varepsilon} + b_3 \frac{\Delta\varepsilon}{\varepsilon^2}\right)\frac{T_i + T_j}{2} + \left(b_4 + b_5 \frac{1-\varepsilon}{\varepsilon} + b_6 \frac{\Delta\varepsilon}{\varepsilon^2}\right)\frac{T_i - T_j}{2} + b_7(T_i - T_j)^2 \tag{2}$$

where $T_i$ and $T_j$ are the brightness temperatures measured in channels $i$ (~11.0 μm) and $j$ (~12.0 μm), respectively; $\varepsilon$ is the

mean emissivity for the two channels ($\varepsilon = 0.5 [\varepsilon_i + \varepsilon_j]$); and $\Delta\varepsilon$ is the emissivity difference between the channels ($\Delta\varepsilon = \varepsilon_i - \varepsilon_j$); $b_k$ ($k = 0, 1, \dots 7$) represents the algorithm coefficients derived from the simulated dataset.

As reported in previous studies (Montanaro et al., 2014a; Barsi et al., 2014; Montanaro et al., 2014b; Montanaro et al., 2014c), thermal infrared radiance measured by Landsat-8 TIRS suffers from straylight, which is caused by out-of-field radiance that scatters onto the detectors, adding a non-uniform banding signal across the field of view. The magnitude of this extra signal can be ~8% or higher (band 11) and is generally twice as large in band 11 as in band 10. Correction algorithms for this artifact have been developed and applied in the new version of level L1T Landsat-8 data (Montanaro et al., 2015), and the straylight artifact in the current product is reduced by half on average (Gerace and Montanaro, 2017). However, the artifact could be amplified in a surface temperature map when SWA is used, with a temperature error of 0~2 K or more (Gerace and Montanaro, 2017), which would certainly impact lead detection from IST maps. A postprocessing procedure utilizing the linear pattern of the straylight artifact is applied to remove this banding noise. First, a median temperature is determined for each image pixel from a long enough along-track-only neighborhood. Then a noise image can be obtained by detrending this median image (Eppler and Full, 1992), thus the surface temperature image from SWA can be improved for lead detection.

## 3.2 Lead detection

In remote sensing images, leads (thin ice and open water) are represented by negative albedo anomalies in the optical range, negative brightness temperature anomalies in near infrared (NIR), and positive surface temperature anomalies compared to the surrounding thick ice (Fett et al., 1997). Variance caused by uneven illumination, view angle, and air temperature should also be taken into account.

Willmes and Heinemann (2015) reported the use of surface temperature anomalies to detect leads. The temperature anomaly $\Delta T_s$ for each IST pixel is defined as a deviation from the median in a square neighborhood, thus temperature variation due to the air temperature field can be removed. This can be expressed in the following equation:

$$\Delta T_s = T_s - M_{T_s,w} \tag{3}$$

where $M_{T_s,w}$ represents the median surface temperature in a square neighborhood with a side length of $w$. This equation was adapted for Landsat-8 IST map using a median from an along-tract-only linear neighborhood to further minimize the straylight artifact. Since median temperature is selected as background temperature, length $w$ should be at least twice as large as the largest lead width within the image area (or along the track) to preserve the lead profile and reduce the temperature gradient caused by air temperature variation across the image.

Generally, surface temperature anomalies for thick ice follow normal distribution with a mean of zero, thus any large deviation from the mean can be assumed as a potential lead and extracted using a proper threshold. Several image-based threshold selection techniques for binary lead segmentation were compared in Willmes and Heinemann (2015), and an iterative threshold selection method (Ridler and Calvard, 1978) was recommended for extracting leads from a temperature anomaly map. Assuming an initial threshold using the mean temperature anomaly ($m_0$) of the whole image, the iterative method seeks a threshold $m_i$ which is the mean of averages $m_A$ and $m_B$ from two clusters divided by $m_i$: Leads (A) and pack ice (B). However, any image-based threshold method provides a threshold that can vary significantly due to temperature noise and lead distribution. For consistency in different scales, several threshold methods were compared for lead detection in both MODIS and TIRS temperature maps (see section 5.1), and iterative threshold method was adopted.

Using width samples crossed by transects, Lindsay and Rothrock (1995) found mean lead width between 2 and 3 km in the Arctic winter; larger means are found in peripheral seas. We modified the method by using an orthogonal system (vertical, south-north; horizontal, west-east; Fig. 2) to determine lead width for every lead pixel. A minimum lead extent in two orthogonal directions was selected for the pixel, i.e., $X = \min(x_1, x_2)$. Since the orientation of a single lead is unknown, this method tends to overestimate width $x$ due to a mismatch between the preset direction and the orientation of the lead (Key and Peckham, 1991), but the orthogonal system will help contain the error ($X \leq \sqrt{2}x$). Since we assign lead width to every pixel across the lead, length $L_i$ for lead width $X_i$ can be calculated as follows:

$$L_i = \frac{a_0^2 N_i}{X_i} = \frac{a_0 N_i}{i} \tag{4}$$

where $a_0$ is the pixel size, for TIRS, the value is 30 m, for MODIS, 1km; and $N_i$ is the number of pixels for width $X_i =$

$a_0i$, ($i = 1, 2, 3…$).

## 3.3 Heat flux model used for lead area

Turbulent heat flux between the Arctic Ocean and the atmosphere, including sensible ($F_s$) and latent ($F_l$) heat flux, is mostly dominated by heat flux over open water and thin ice, which constitute leads in pack ice and polynya in coastal area. Turbulent heat flux over leads can be estimated using bulk aerodynamic formulae or a fetch-limited model developed based on field observations.

### 3.3.1 Bulk aerodynamic formulae

Assuming that temperatures in the atmospheric boundary layer are determined by the heat balance over thicker ice and turbulent heat exchange does not vary significantly across the narrow areas of leads, then turbulent heat fluxes are mainly determined by temperature and humidity differences between the surface and atmosphere at reference height $r$ (Maykut, 1978). Sensible heat flux ($F_s$) and latent heat flux ($F_l$) are given by the following bulk formulae:

$$F_s = \rho_a c_p C_{sh} u_r (T_s - T_r) \tag{5}$$
$$F_l = \rho_a L_v C_{le} u_r (Q_s - Q_r) \tag{6}$$

where $\rho_a$ is the air density; $c_p$ is the specific heat at constant pressure; $L_v$ is the latent heat of vaporization; $u_r$, $T_r$, and $Q_r$ are wind speed, air temperature, and specific humidity at reference height $r = 2$ m, respectively; $T_s$ is surface temperature; and $Q_s$ is specific humidity close to the surface. Assuming that air at the surface of ice or water is always saturated, the specific humidity at the surface can be derived as:

$$Q_s = \frac{0.622 e_{s0}}{p - 0.378 e_{s0}} \tag{7}$$

where $p$ is the air pressure, and $e_{s0}$ represents the saturated vapor pressure at surface temperature $T_s$:

$$e_{s0} = e_0 10^{\frac{at}{b+t}} \tag{8}$$

with $e_0$ representing saturated vapor pressure at 0 °C, approximately 6.11 hPa; $t$ is temperature in Celsius; and $a$ and $b$ are coefficients (for water surface, $a = 7.5$, $b = 237.3$ K; for thin ice, $a = 9.5$, $b = 265.5$ K). These equations are also applied for specific humidity at 2 m height using dew point temperature data from ERA-interim.

$C_{sh}$ and $C_{le}$ are transfer coefficients for sensible heat and latent heat, calculated using equations from Oberhuber (1988) and Goosse et al. (2000) (see Appendix B). Since the wind speed and air temperature from ERA-interim are at different heights, a wind profile equation was used to calculate wind speed at 2 m height (Ray et al., 2006):

$$\frac{u_{10}}{u_r} = \frac{\ln 10 - \ln Z_0}{\ln r - \ln Z_0} \tag{9}$$

where $u_{10}$ and $u_r$ are wind speed at 10 m and 2 m height, and $Z_0$ is surface roughness length. In our study area, the main direction of wind from the reanalysis dataset is roughly perpendicular to the dominant orientation of leads. Therefore, only the wind magnitude was used in our study.

### 3.3.2 Fetch-limited model

When cold air travels to a warmer surface, a convective atmospheric TIBL forms and thickens with distance downwind of the surface discontinuity or fetch $X$ (Stull, 1988). As the wind blows over water (or thin ice), the near-surface air gets warmer with more vapor, while new ice accumulates at the downwind side of the lead, progressively narrows, and seals the window. Thus, the temperature and humidity differences between the air and the surface decrease. Since sensible and latent heat fluxes are proportional to temperature and humidity differences, respectively, turbulent heat transfer also recedes with increasing lead width or fetch. Another mechanism is described in Esau (2007) for leads 1–10 km wide. Under weak wind conditions (~2 m/s), convective overturning prevents cold breezes from penetrating into the lead area, reducing the average turbulent heat flux.

To estimate turbulent heat flux over small leads, fetch-limited models were developed based on a few observations (Andreas and Murphy, 1986; Alam and Curry; 1997; Andreas and Cash, 1999). However, the assumption of universal water

surface in leads and the application of sea surface roughness model (Andreas and Murphy, 1986; Alam and Curry, 1997) are not applicable in our case, where open water and thin ice dominate. Since the signal of TIBL is absent in the coarse grid of 2 m air temperature from the ERA reanalysis dataset, the data might not be appropriate to demonstrate the Alam and Curry (1997) model, which relies on accurate measurement of meteorological parameters. Whereas the Andreas and Cash (1999) model is more sensitive to lead width than atmospheric conditions (Marcq and Weiss, 2012). Therefore, only the Andreas and Cash (1999) model was used in our experiment.

Andreas and Cash (1999) gave direct formulations of heat fluxes as a function of lead width $X$ based on data fitting from three sets of measurements. For free convection conditions in large fetch:

$$F_{s(X)} = C_* \rho_a C_p D (T_s - T_r)/\Delta z_T \tag{10}$$

$$F_{l(X)} = C_* \rho_a L_v D_w (Q_s - Q_r)/\Delta z_Q \tag{11}$$

where $D$ and $D_w$ are the molecular diffusivities of heat and water vapor in air, respectively, and $\Delta z_T$ and $\Delta z_Q$ are length scales for heat and humidity, respectively, which consider the viscosity of air $v$ and buoyancy differences between the surface and reference height $r$:

$$\Delta z_T = \left(\frac{vD}{\Delta B}\right)^{1/3} \tag{12}$$

$$\Delta z_Q = \left(\frac{vD_w}{\Delta B}\right)^{1/3} \tag{13}$$

$$\Delta B = \frac{g}{\bar{T}}\left(\Delta T + \frac{0.61\bar{T}\Delta Q}{1+0.61\bar{Q}}\right) \tag{14}$$

where $\Delta B$ is the buoyancy difference; $g$ is acceleration due to gravity; $\Delta T$ and $\Delta Q$ are the difference of temperature and specific humidity between surface and air at reference height r, respectively; and $\bar{T}$ and $\bar{Q}$ are the average temperature and specific humidity between them.

The coefficient $C_*$ is a function of stability, which facilitates the generalization of Eq. (10) and (11) to the transition between free and forced convection, thus making them applicable to smaller fetch. $C_*$ is estimated using lead and polynya data:

$$C_* = \frac{0.3}{0.4-h/L} + 0.15 \tag{15}$$

$$h = 0.82 \ln X + 0.02 \tag{16}$$

where $h$ is the depth of the TIBL in meters as a function of lead width $X$, and $L$ is the Obukhov length given in Eq. (17); $L$ is a length scale of stability and is negative for unstable stratification, while its magnitude rises with instability.

$$L^{-1} = 8.0 * \left(\frac{0.65}{r} + 0.079 - 0.0043r\right)*Ri_b \tag{17}$$

where $Ri_b$ is the bulk Richardson number:

$$Ri_b = -\frac{r g}{\bar{T}}\frac{T_s - T_r}{u_r^2} \tag{18}$$

where $u_r$ is wind speed obtained from Eq. (9). Apart from lead width, meteorological parameters are also important in the model. As shown in Fig. 3, for the narrowest lead from TIRS ($X = 30$ m), turbulent heat flux, especially sensible heat, rises quickly with larger temperature difference and stronger wind. Most importantly, assuming a constant temperature difference and steady crossing wind, heat flux decreases with increasing fetch and becomes saturated for lead width great than 1 km, as depicted in Fig. 4. As the fetch dependence of heat flux over lead is negligible for lead widths greater than 1 km, this model was applied to TIRS data only.

## 4 Results

### 4.1 Ice surface temperature

IST maps retrieved from MODIS and TIRS using Eq. (1) and (2) are shown in Fig. 5. The temperature signature of small leads

in the northern part of the image area is largely reduced in the MODIS IST map, due to its coarse resolution and heterogeneous pixels, compared to that from TIRS. In addition, the banding effect of straylight is very obvious in the TIRS IST map. This artifact was detected and removed by using a median from the along-track linear neighborhood and detrending the median image (Fig. 6). The corrected TIRS IST map is shown in Fig. 5 for comparison.

Although the median and artifact images show a little bias around large leads, the corrected TIRS IST map is very smooth and more suitable for lead detection and heat flux calculation. Scatter plots of IST from MODIS and TIRS before and after correction are shown in Fig. 7. The correlation of IST from two sensors estimated by interpolating MODIS IST to the TIRS scale (30 m) is quite good, with a Pearson coefficient of approximately 0.9 (0.902 and 0.896 before and after correction for straylight, respectively). The primary coefficient of linear regression improved from 1.025 to 1.004 before and after correction, indicating that the corrected TIRS IST is in better agreement with MODIS. However, the root mean square error (RMSE) from regressions increased from 1.216 K to 1.233 K. It also reveals that for the 250–270 K temperature range, IST retrieved from TIRS is generally 0.61–0.70 K higher than that from MODIS.

## 4.2 Sea ice lead retrieval

Regional temperature anomaly maps calculated from IST maps are shown in Fig. 8. The mean surface temperature anomaly is 0.116 K with a standard deviation (Std) of 1.180 K for MODIS, and 0.283 K with a Std of 1.619 K for TIRS.

Binary lead maps were generated using iterative thresholds (Fig. 9). Large floes and small leads dominate the northern part of the images, where temperature is lower, while two very large leads can be observed in the southern portion. The maps illustrate that the lead binary retrieved from MODIS captures major lead structures, but small leads are missed in most cases compared to leads detected from TIRS.

Lead area estimated from MODIS is 8074.0 km$^2$, which accounts for 8.25% of the frame area, and for TIRS, 7376.2 km$^2$ and 7.53%. Validation with Landsat-8 panchromatic images shows that large leads tend to be amplified by blurred mixed pixels along boundaries, while small leads are neglected due to the coarse resolution of MODIS.

Lead width was calculated for every lead pixel in the binary maps from MODIS and TIRS, and divided into three categories (Table 2): small leads (width ≤ 1 km), medium leads (1 km < width ≤ 5 km), and large leads (width > 5km). Although the 1 km resolution is the finest for MODIS thermal, the 1 km wide lead category should provide a reasonable guess of potential small leads or subpixel leads at MODIS scale (Lindsay and Rothrock, 1995).

The width distribution of leads from MODIS and small leads from TIRS are plotted in Fig. 10 relating to the lengths of leads. Similar to the concept of number density, the length of each lead width can be fitted with a power law distribution, and the exponents from power law fitting are 2.241 and 2.346 for leads from MODIS and TIRS, respectively. The power law distribution indicates that narrow leads are prevalent, while a larger exponent implies that smaller leads are more dominant at TIRS scale.

The total length of leads with various widths is 10150.3 km from TIRS, including 8502.2 km (83.76%) from small leads with width no more than 1 km, compared to a total length of 2746.4 km from MODIS, where the narrow leads (1 km wide) only account for 1050.0 km (38.23%). Total length of leads is underestimated by 72.9% in MODIS data compared to TIRS data. As for the area of leads, small leads (width ≤ 1 km) account for 34.54% of total lead area from TIRS and only 13.00% of lead area from MODIS (Table 2).

## 4.3 Heat flux over leads

IST, described in Section 4.1, and lead width from TIRS (Section 4.2) were used in bulk formulae and the fetch-limited model along with ERA-interim reanalysis data to estimate turbulent heat flux through leads. For consistency, the estimated heat flux is positive from ocean to atmosphere.

### 4.3.1 Bulk formulae

Turbulent heat flux over lead area is obtained by summing up sensible and latent heat flux from Eq. (5) and (6) using IST and lead maps retrieved from MODIS or TIRS (Fig. 11). Table 2 reveals that total heat flux over leads calculated using TIRS IST

is $8.40 \times 10^{11}$ W over a total area of 7376.2 km$^2$. This is 56.70% larger than that from MODIS data ($5.36 \times 10^{11}$ W). About 23% of the difference can be explained by IST bias between MODIS and TIRS, but most of the difference comes from small leads. Small leads account for $2.16 \times 10^{11}$ W (25.75%) of total heat flux in TIRS data, almost seven times the heat flux from the narrow lead category in MODIS ($3.10 \times 10^{10}$ W, 5.79%).

**4.3.2 The Andreas and Cash (1999) model**

As we can see in Fig. 11 and Table 3, total heat flux over leads estimated by the Andreas and Cash (1999) model is $1.11 \times 10^{12}$ W, 32.34% higher than that from bulk formulae, i.e. $8.40 \times 10^{11}$ W, among which 32.95% of the difference comes from the small lead class. In both cases, small leads account for a quarter or more of total heat flux over all leads in both models, due to the large area, though the heat flux estimated using the fetch-limited model is $3.06 \times 10^{11}$ W, 41.39% larger than the $2.16 \times 10^{11}$ W from bulk formulae. For comparison, the estimated heat fluxes from medium and large lead classes also increased by 38.95% and 28.10%, respectively, when the Andreas and Cash (1999) model was applied. However, the contribution of turbulent heat flux from large leads is reduced from 34.17% to 32.68%, while the contribution from small leads increased from 25.75% to 27.50%. Nonetheless, the fact that large leads with widths greater than 5 km account for 27.16% of total lead area but contribute more than 32% of total heat flux over leads is somehow contradictory to the fetch-limited theory.

Inspection of input data revealed that the 2 m air temperature from ERA-interim has almost the same mean value around 262 K as the surface temperature from Landsat-8. The temperature difference between air and surface, $\Delta T$, spreads from 1.58 to 12.38 K, with a mean of 4.88 K, along with an average wind speed of about 7 m/s at 2 m height over leads. The distributions of air temperature and surface temperature of leads are plotted in Fig. 12. The temperature difference over narrow leads is too small to obtain a robust estimation of turbulent heat flux.

**5 Discussion**

**5.1 Threshold method**

The operational definition of a lead is a fracture or passageway through ice that is navigable by surface vessels (Canadian Ice Service, 2005; World Meteorological Organization, 2014). However, within any optical, thermal, or microwave image, the radiometric signature of a narrow lead with open water may be identical to that of a wider lead with thin ice. In most studies involving the utility of remote sensing data, any linear features of open water or thin ice within pack ice are regarded as leads for convenience (Fetterer and Holyer, 1989; Fily and Rothrock 1990; Lindsay and Rothrock, 1995). Due to the confusion in the definition of leads in remote sensing images and the need to extract lead signatures from the background, threshold segmentation has been frequently used (Eppler and Full, 1992; Lindsay and Rothrock, 1995; Weiss and Marsan, 2004; Marcq and Weiss, 2012). Willmes and Heinemann (2015) presented several threshold selection techniques for binary lead segmentation. However, thresholds given by image-based methods can vary significantly depending on noise level (caused by air temperature variance) and lead distribution.

In our study, a set of thresholds was tested for extracting leads from temperature anomaly maps, areal fractions of leads from fixed thresholds, Std thresholds, and an iterative threshold are shown in Table 4. The obtained lead fractions are a composite of thresholds and contrast in surface temperature of leads compared to the surrounding ice, i.e., temperature anomaly $\Delta T_S$. Since the anomaly maps from the two sensors have different means and standard deviations, mainly due to different definitions of neighborhood in calculating $\Delta T_S$, the results from a fixed threshold might be biased. The iterative thresholds from both anomaly maps are a little larger than their first Std thresholds. The difference in lead fractions from the two sensors mainly resulted from mixed pixels at MODIS scale, but the threshold should also be considered. When high thresholds (2nd and 3rd Std) are applied, the lead fraction extracted from MODIS drops quickly below that from TIRS (as shown in Table 4), and this is consistent with results from Key et al. (1994). While larger thresholds lead to underestimating lead distribution, lower thresholds allow more pixels to be detected as leads, also giving rise to false leads caused by air temperature variance.

Validation with Landsat-8 panchromatic images shows that the iterative threshold detects most lead structures (89.5%) and exhibited better resistance against air temperature noise. Thus, iterative thresholds were selected for lead extraction in this

study.

## 5.2 Lead width

Lead geometry and distribution in the Arctic have been studied using optical and microwave remote sensing data (Fily and Rothrock, 1990; Lindsay and Rothrock, 1995; Tschudi et al., 2002). A simple one-parameter exponential model was used for number density distribution of lead width (Key and Peckham, 1991; Key et al., 1994; Maslanik and Key, 1995):

$$f_{(X)} = \frac{1}{\lambda} e^{\frac{-X}{\lambda}} \qquad (19)$$

where $\lambda$ is the mean lead width. However, a mean lead width can be oversimplified in diverse circumstances. Lindsay and Rothrock (1995) reported the power law distribution of lead width in AVHRR imagery:

$$N_{T(X)} = aX^{-b} \qquad (20)$$

where $N_{T(w)}$ is the number density of leads of width $w$ per kilometer of width increment; the exponent $b$ indicates the relative frequency of large and small leads, while the coefficient $a$ is directly related to the lead concentration and the range of widths over which the power law is thought to apply. The annual mean of exponent $b$ was found to be 1.60 using AVHRR images (Lindsay and Rothrock, 1995). Larger values of $b$ were reported using data with better resolution: 2 and 2.29 for submarine sonar observation in Fram Strait (Wadhams, 1981) and Davis Strait (Wadhams et al, 1985) when a 100 m bin width was used, 2.1–2.6 for 20 m SPOT images in orthographic directions using different thresholds (Marcq and Weiss, 2012). Note that most of these studies used only width samples crossed by limited linear transects.

In our study, although lead width follows the power law distribution at both scales, the fitted exponents vary from 2.241 to 2.346 at resolution from 30 m to 1 km. Since the 30 m L1T images were resampled from the original 100 m TIRS data, the actual distribution of leads less than 100 m wide is debatable. In comparison with Landsat-8 TIRS and panchromatic images, we find that the lead map generated from the MODIS IST data neglects very small leads, but overestimates the width of other leads approximately 1 km wide. Overall, the 1 km wide lead category at MODIS scale should provide a reasonable guess of potential small or subpixel leads. The small leads retrieved using TIRS provide a valuable reference for the capacity of MODIS to detect narrow leads.

## 5.3 Comparison of the models

In both the Andreas and Murphy (1986) and Andreas and Cash (1999) models, for reference height r < 10 m, the ratio between roughness lengths for momentum and heat, $Z_0/Z_T$, is assumed to be ~$e^2$ to calculate Obukhov length $L$ using Richardson number $Ri_b$ (see Eq. (17)). The calculated Obukhov length $L$ has absolute values about 67% higher than those using Eq. (B8) and (B13) from the bulk formulae (Oberhuber, 1988; Goosse et al., 2000). If Eq. (B8) and (B13) were used to solve Obukhov length $L$ and coefficient $C_*$ in the Andreas and Cash (1999) model, estimated turbulent heat flux would be smaller (Table 3), but still 15.53% larger than that from the bulk formulae, with an even larger part of the difference from the small lead category (42.48%, compared to 32.96% in Section 4.3.2).

Our results suggest that the contribution of heat flux from small leads mainly results from their large length, or number density, and vast area instead of efficiency. Though small leads are more efficient for heat exchange between the ocean and the atmosphere, thin ice growing in newly opened leads can quickly cover the exposed ocean surface, thus reducing heat exchange. Moreover, due to the mixture of subpixel leads and thick ice, the surface temperature of some pixels in small leads is much lower than the freezing point.

Nonetheless, our results show that the fetch-limited model could be used to estimate turbulent heat flux on a regional scale with surface temperature fields from remote sensing. However, the fetch-limited model proposed by Andreas and Cash (1999) was based mainly on a few observations over open leads and polynya, while most lead pixels detected using temperature anomalies in our study are likely covered by thin ice (surface temperature <270 K, Fig. 12). Thus, near-surface air temperature with finer resolution is needed for validating the turbulent heat flux estimated using the fetch-limited model.

## 5.4 Heat flux for larger temperature differences

For comparison, a test using preset meteorological conditions was performed using the TIRS lead binary. Assuming the surface temperature in leads is right at the freezing point, with a wind speed of 7 m/s at 2 m height and a temperature difference of 5 K and 10 K, turbulent heat fluxes from both models were calculated (Fig. 13), and are summarized in Table 5. Note that lead width in Fig. 13 is on a logarithmic scale.

Clearly, turbulent heat flux estimated using the Andreas and Cash (1999) model is always higher than that using the bulk formulae. For both models, estimated turbulent heat flux with $\Delta T$ of 5 K or 10 K peaks at lead width of ~270 m, a smaller width than the 360 m using $\Delta T$ obtained from TIRS IST and air temperature from ERA-interim.

The distribution of turbulent heat flux estimated using bulk formulae with $\Delta T$ of 5 K and 10 K depends on the areal fraction from each lead category. The contribution from leads with widths greater than 1 km converges to the lower end with fluctuation. As expected, the estimated total heat flux of $1.68 \times 10^{12}$ W at $\Delta T = 10$ K is about twice as large as that at $\Delta T = 5$ K ($8.46 \times 10^{11}$ W).

When the Andreas and Cash (1999) model was applied, small leads were found to have a larger contribution at higher $\Delta T$, $3.27 \times 10^{11}$ W (35.86%) and $6.66 \times 10^{11}$ W (36.57%) at $\Delta T = 5$ K and 10 K, respectively, compared to the areal fraction of 34.54%. More contributions from small leads can be expected at larger temperature differences and stronger wind in winter.

## 6 Conclusions

Although the same local temperature anomaly and threshold methods were applied, leads retrieved at MODIS and Landsat-8 TIRS resolution scales presented very different geometries and distributions. Within the studied area, the total length of leads is 10,150.3 km from TIRS, including 8502.2 km (83.76%) from small leads with width less than 1 km. This is in contrast to the total length of 2746.4 km from MODIS, where the narrow leads (1 km wide) only account for 1050.0 km (38.23%). The total length of leads is underestimated by 72.9% in the MODIS data. For the area of leads, small leads (width $\leq$ 1km) account for 34.54% of the total lead area from TIRS, but only 13.00% of the total lead area from MODIS. Although the lead width follows the power law distribution at both scales, the fitted exponents vary from 2.241 to 2.346.

When bulk aerodynamic formulae are applied to the reanalysis dataset, the heat flux estimated using TIRS data is $8.40 \times 10^{11}$ W, 56.70% larger than that from MODIS data ($5.36 \times 10^{11}$ W). About 23% of the difference can be explained by IST bias between MODIS and TIRS, but most of the difference comes from small leads. Small leads account for $2.16 \times 10^{11}$ W (25.75%) of the total heat flux over all leads in the TIRS data, almost seven times the heat flux from the narrow lead category in MODIS ($3.10 \times 10^{10}$ W, 5.79%).

The turbulent heat flux over leads estimated from the TIRS data by the Andreas and Cash (1999) model is $1.11 \times 10^{12}$ W, 32.34% higher than that from bulk formulae ($8.40 \times 10^{11}$ W). In both cases, small leads account for about a quarter of the total heat flux in both models, due to the large area, though the heat flux estimated using the fetch-limited model is 41.39% larger. A greater contribution from small leads can be expected with larger temperature differences and stronger wind conditions. A near-surface air temperature with finer resolution is still needed for validation of turbulent heat flux estimated using the fetch-limited model before extensive application.

*Author contributions.* Xiaoping Pang and Xi Zhao designed the experiments and Meng Qu carried them out. Jinlun Zhang provided valuable instructions on data acquisition and manuscript editing. Qing Ji and Pei Fan helped to develop the model code. Meng Qu prepared the manuscript with contributions from all co-authors.

*Competing interests.* The authors declare that they have no conflict of interest.

*Acknowledgements.* This work was supported by the National Natural Science Foundation of China (Nos. 41876223, 41576188, and 41606215) and the National Key Research and Development Program of China (2016YFC1402704). Jinlun Zhang was supported by NOAA Climate Program Office (NA15OAR4310170). The authors acknowledge the NASA Goddard Space

Flight Center, the U.S. Geological Survey (USGS), and the European Center for Medium-Range Weather Forecasts (ECMWF) for providing the images and datasets used in this study. Finally, we would like to express our thanks to the anonymous reviewers and handling editor Christian Haas for their valuable comments which helped improve our manuscript.

**Appendix A**

5 **Validation using Landsat-8 panchromatic images**

Top of atmosphere (TOA) reflectance from Landsat-8 panchromatic images were corrected for solar zenith angle and mosaicked for validation. Using Jenks's natural breaks classification method (Jenks, 1963), panchromatic pixels were classified into three surface categories: open water and thin ice, refrozen leads, and pack ice. In terms of turbulent heat flux, only pixels in the open water and thin ice category were regarded as leads. As can be seen in Table A1, the producer's accuracy
10 of lead detection using the iterative threshold is 89.5%, with an omission error of 10.5% and a commission error of 16.1%.

**Appendix B**

Equations used for turbulent heat flux estimation using bulk formulae (Large and Pond, 1981, 1982; Oberhuber, 1988; Goosse et al, 2000; Marcq and Weiss, 2012) are as follows:

$$c_{sh} = 0.0327 \frac{k}{\ln(r/z_0)} \Phi_{sh} = c_{shN}\Phi_{sh} \tag{B1}$$

$$c_{le} = 0.0346 \frac{k}{\ln(r/z_0)} \Phi_{le} = c_{leN}\Phi_{le} \tag{B2}$$

$$\Phi_{sh} = \frac{\sqrt{c_M/c_{MN}}}{1 - c_{shN}k^{-1}C_{MN}^{-1/2}\Psi_H} \tag{B3}$$

$$\Phi_{le} = \frac{\sqrt{c_M/c_{MN}}}{1 - c_{leN}k^{-1}C_{MN}^{-1/2}\Psi_L} \tag{B4}$$

$$\sqrt{\frac{c_M}{c_{MN}}} = \frac{1}{(1 - \sqrt{c_{MN}}k^{-1}\Psi_M)} \tag{B5}$$

$$c_{MN} = \frac{k^2}{\left(\ln\left(\frac{r}{z_0}\right)\right)^2} \tag{B6}$$

$$u_*^2 = c_M u_r^2 \tag{B7}$$
$$T_0 = T_r(1 + 2.2 \times 10^{-3} T_r Q_r) \tag{B8}$$

Surface roughness lengths for momentum are given as:

$$z_0 = 0.032 \frac{u_*^2}{g} \tag{B9}$$

For unstable conditions:

25
$$\Psi_H(A) = \Psi_L(A) = 2\ln\left(\frac{1+A^2}{2}\right) \tag{B10}$$

$$\Psi_M(A) = 2\ln\left(\frac{1+A}{2}\right) + \ln\left(\frac{1+A^2}{2}\right) - 2\arctan A + \frac{\pi}{2} \tag{B11}$$

$$A = \left(1 - 16(r/L)\right)^{1/4} \tag{B12}$$

$$r/L = \frac{100r}{T_0 u_r^2}\left((T_s - T_r) + 2.2 \times 10^{-3} T_0^2 (Q_s - Q_r)\right) \tag{B13}$$

**Appendix C**

**Constants**

Constants used in IST calculation from Landsat-8 TIRS (Du et al. 2015) are as follows:

1. ASTER emissivity library (Skoković et al., 2014):

   $\varepsilon_{water,10} = 0.991$; $\varepsilon_{water,11} = 0.986$; $\varepsilon_{snow/ice,10} = 0.986$; $\varepsilon_{snow/ice,11} = 0.959$

   $\bar{\varepsilon}_{water} = 0.9885$; $\Delta\varepsilon_{water} = 0.005$

   $\bar{\varepsilon}_{snow/ice} = 0.9725$; $\Delta\varepsilon_{snow/ice} = 0.027$

2. NIR reflectance threshold for classification between water and ice/snow: 0.1
3. Water vapor content from MOD05: $< 2.5$ g·cm$^{-2}$
4. $b_i$: $b_{0\sim7}$: [–2.78009, 1.01408, 0.15833, –3.4991, 4.04487, 3.55414, –8.88394, 0.09152]
5. RMSE: 0.34 K

Constants used in turbulent heat flux estimation:

Air pressure: $p = 101$ kPa

Air density: $\rho_a = 1.3$ kg m$^{-3}$

Kinematic viscosity of air: $v = 1.31 \times 10^{-5}$ m$^2$·s$^{-1}$

Molecular diffusivities of heat in the air: $D = 1.86 \times 10^{-5}$ m$^2$ s$^{-1}$

Molecular diffusivities of water vapor in the air: $D_w = 2.14 \times 10^{-5}$ m$^2$ s$^{-1}$

Specific heat at constant pressure: $c_p = 1004$ J kg$^{-1}$ K$^{-1}$

Latent heat of vaporization or sublimation: $L_{water} = 2.51 \times 10^6$ J·kg$^{-1}$, $L_{ice} = 2.86 \times 10^6$ J·kg$^{-1}$

Reference height: $r = 2$ m

Von Karman constant: $k = 0.4$

Gravitational constant: $g = 9.8$ m s$^{-2}$

Salinity of sea water in the Beaufort Sea: $S_w = 27.947$ (‰)

Freezing point of sea water:

$T_{S0} = 273.15 - 0.0137 - 0.05199 S_w - 0.00007225 S_w^2 = 271.68$ K

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

**Figure Captions**

15    **Figure 1.** Location of study area. Background image is brightness temperature from Moderate Resolution Imaging Spectroradiometer (MODIS) band 31 (~11 μm). Location of Landsat-8 images is marked by a red rectangle.

**Figure 2.** Detection of lead width using two orthogonal directions. $x$ is the real lead width. Lead extents in orthogonal system in $v$ and $h$ directions are measured as $x_1$ and $x_2$, respectively.

**Figure 3.** Turbulent heat flux rises with increasing temperature difference $\Delta T$ and intense wind at lead width of 30 m. Solid and dashed lines represent sensible and latent heat, respectively. Wind speed is illustrated by line color. Clearly, sensible heat flux is basically proportional to $\Delta T$.

**Figure 4.** Turbulent heat flux for each width at wind speed of 5 m/s. Temperature difference between air and lead surface is marked by line color.

**Figure 5.** Ice surface temperature (IST) maps from MODIS and Landsat-8 Thermal Infrared Sensor (TIRS) using split-window algorithms: (**a**) IST map from MODIS; (**b**) IST map from Landsat-8 TIRS; (**c**) corrected IST map from TIRS.

**Figure 6.** Local median and noise image from TIRS IST: (**a**) along-track median temperature map; (**b**) noise image by detrending of median temperature map.

**Figure 7.** Correlation between IST from MODIS and Landsat-8 TIRS before and after correction for straylight. Black lines are reference for x = y, red lines are linear regression lines with a fitting equation. Number density of scattered points is marked by color. (**a**) Scatter plot of IST from MODIS and Landsat-8 TIRS using split-window algorithm; (**b**) scatter plot of IST from MODIS and corrected IST from Landsat-8 TIRS.

**Figure 8.** Local temperature anomalies from (**a**) MODIS and (**b**) Landsat-8 TIRS.

**Figure 9.** Binary lead maps from (**a**) MODIS and (**b**) Landsat-8 TIRS.

**Figure 10.** Width distribution of leads from MODIS and TIRS in log-log plot. Data points from MODIS and TIRS are plotted as orange and blue dots, respectively. Power law fitting is applied. Fitting equations and R squares are shown for comparison.

**Figure 11.** Spatial distribution of heat flux derived from MODIS and Landsat-8 using bulk formulae and fetch-limited model. (**a**) Turbulent heat flux from MODIS using bulk formulae; (**b**) turbulent heat flux from Landsat-8 TIRS using bulk formulae; (**c**) turbulent heat flux from Landsat-8 TIRS using fetch-limited model.

**Figure 12.** Distribution of 2 m air temperature over leads and surface temperature of all leads; small leads with width <1 km, larger leads with width <5 km.

**Figure 13.** Contribution of heat flux from each lead width using bulk formulae and fetch-limited model. Turbulent heat flux retrieved using fetch-limited model and bulk formulae are plotted using solid and dashed lines, respectively. Heat flux calculated using satellite surface temperature, air temperature, and wind speed from reanalysis datasets is drawn in orange; simulated heat flux at $\Delta T = 5$ K and 10 K is in blue and green, respectively.

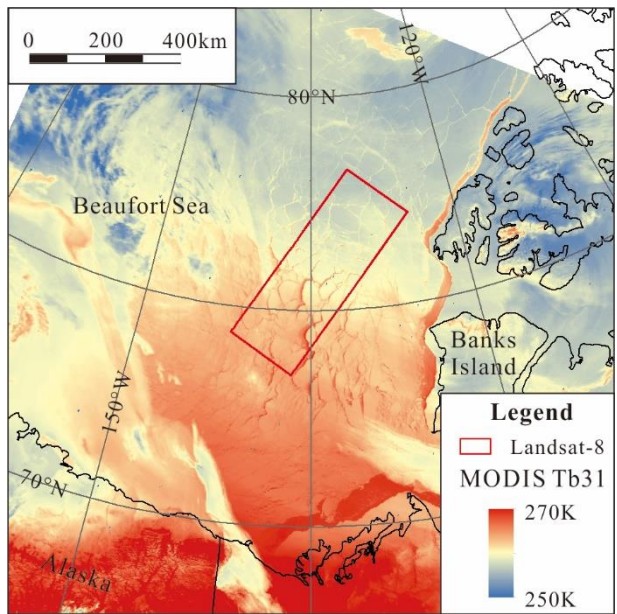

**Figure 1.** Location of study area. Background image is brightness temperature from Moderate Resolution Imaging Spectroradiometer (MODIS) band 31 (~11 μm). Location of Landsat-8 images is marked by a red rectangle.

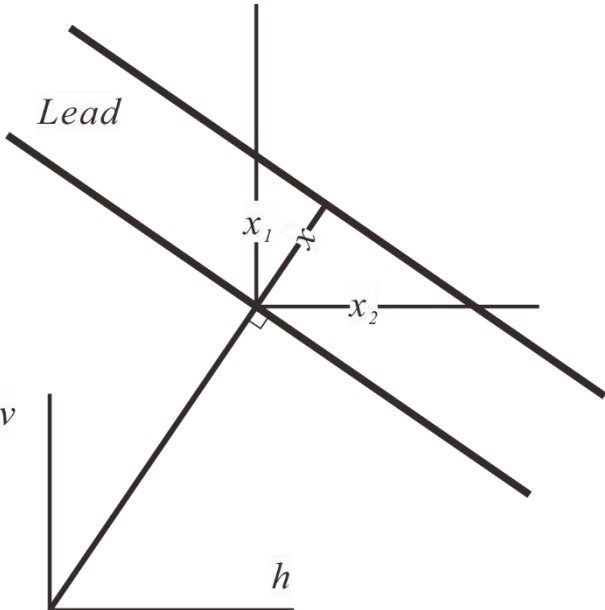

5    **Figure 2.** Detection of lead width using two orthogonal directions. $x$ is the real lead width. Lead extents in orthogonal system in $v$ and $h$ directions are measured as $x_1$ and $x_2$, respectively.

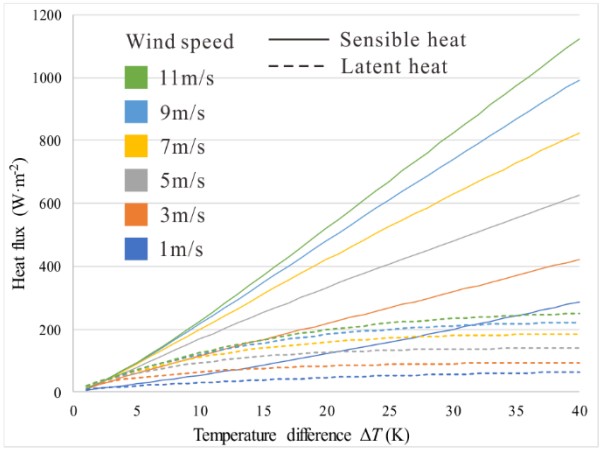

**Figure 3.** Turbulent heat flux rises with increasing temperature difference $\Delta T$ and intense wind at lead width of 30 m. Solid and dashed lines represent sensible and latent heat, respectively. Wind speed is illustrated by line color. Clearly, sensible heat

flux is basically proportional to $\Delta T$.

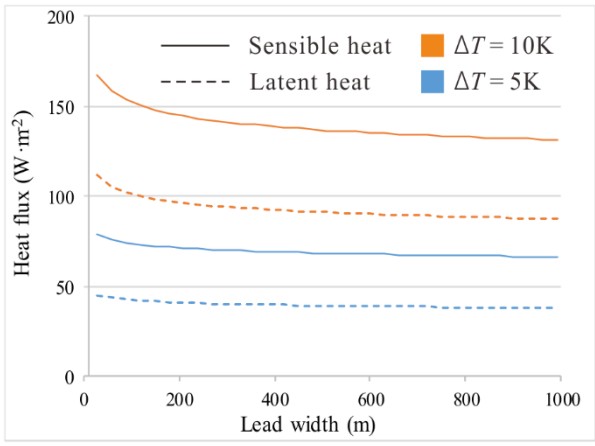

**Figure 4.** Turbulent heat flux for each width at wind speed of 5 m/s. Temperature difference between air and lead surface is marked by line color.

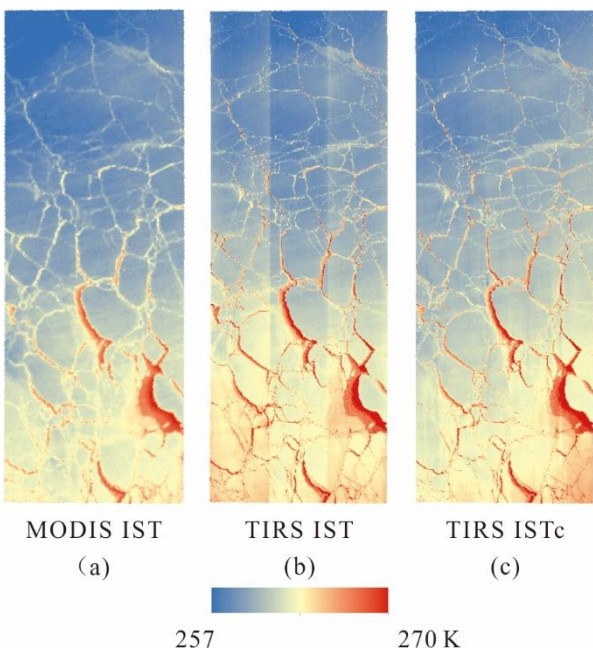

MODIS IST  TIRS IST  TIRS ISTc

(a)  (b)  (c)

257  270 K

**Figure 5.** Ice surface temperature (IST) maps from MODIS and Landsat-8 Thermal Infrared Sensor (TIRS) using split-window algorithms: (**a**) IST map from MODIS; (**b**) IST map from Landsat-8 TIRS; (**c**) corrected IST map from TIRS.

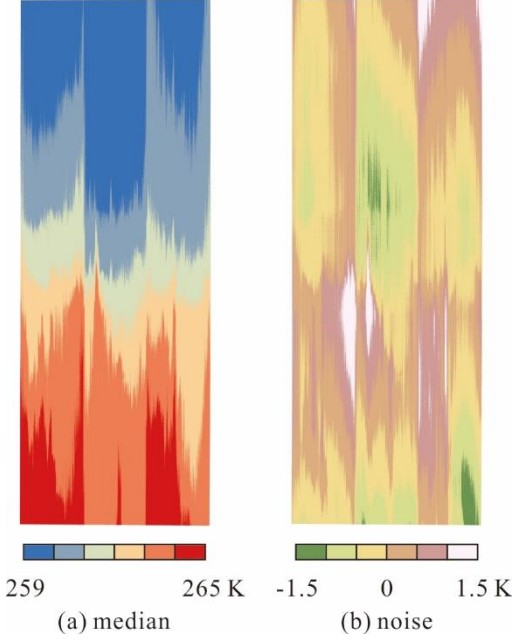

259  265 K  -1.5  0  1.5 K

(a) median  (b) noise

**Figure 6.** Local median and noise image from TIRS IST: (**a**) along-track median temperature map; (**b**) noise image by detrending of median temperature map.

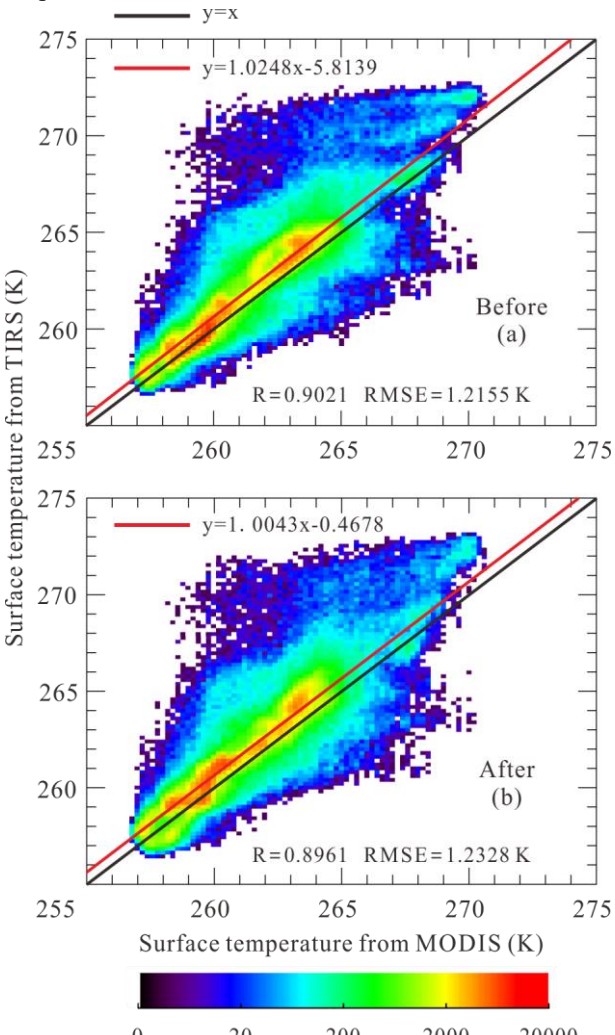

**Figure 7.** Correlation between IST from MODIS and Landsat-8 TIRS before and after correction for straylight. Black lines are reference for x = y, red lines are linear regression lines with a fitting equation. Number density of scattered points is marked by color. (**a**) Scatter plot of IST from MODIS and Landsat-8 TIRS using split-window algorithm; (**b**) scatter plot of IST from MODIS and corrected IST from Landsat-8 TIRS.

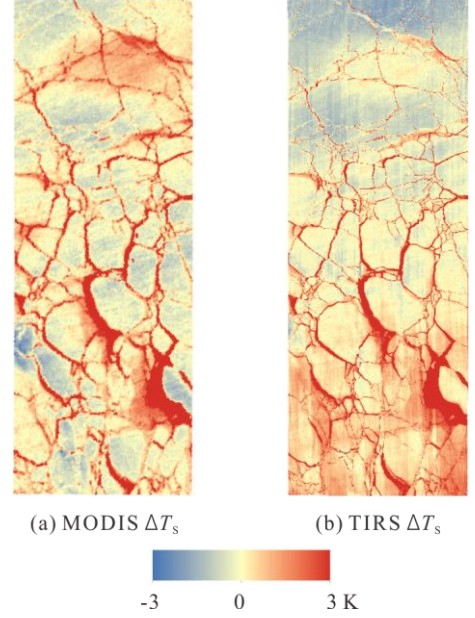

(a) MODIS $\Delta T_s$      (b) TIRS $\Delta T_s$

**Figure 8.** Local temperature anomalies from (**a**) MODIS and (**b**) Landsat-8 TIRS.

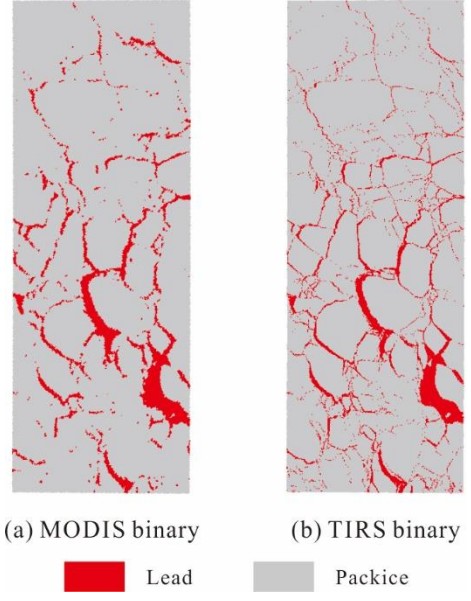

(a) MODIS binary      (b) TIRS binary

■ Lead      ■ Packice

**Figure 9.** Binary lead maps from (**a**) MODIS and (**b**) Landsat-8 TIRS.

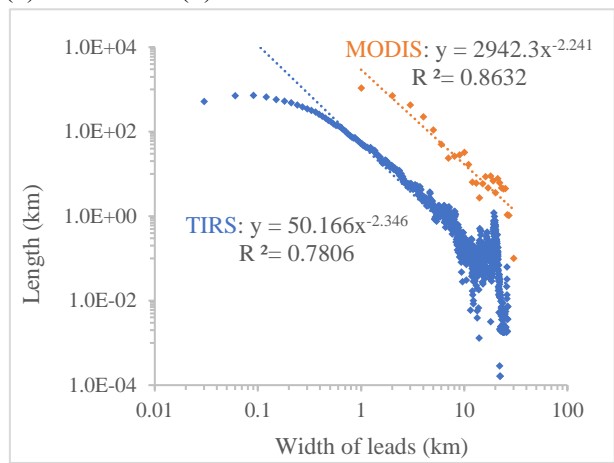

**Figure 10.** Width distribution of leads from MODIS and TIRS in log-log plot. Data points from MODIS and TIRS are plotted
as orange and blue dots, respectively. Power law fitting is applied. Fitting equations and R squares are shown for comparison.

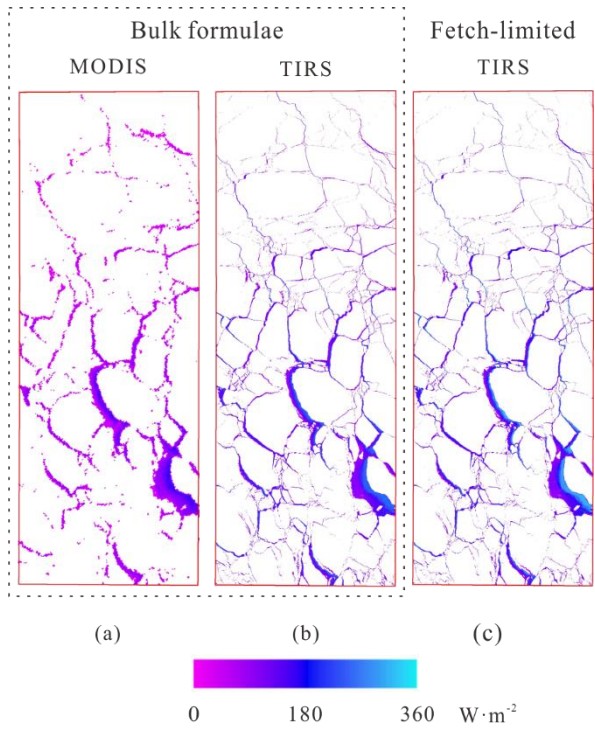

**Figure 11.** Spatial distribution of heat flux derived from MODIS and Landsat-8 using bulk formulae and fetch-limited model. (**a**) Turbulent heat flux from MODIS using bulk formulae; (**b**) turbulent heat flux from Landsat-8 TIRS using bulk formulae; (**c**) turbulent heat flux from Landsat-8 TIRS using fetch-limited model.

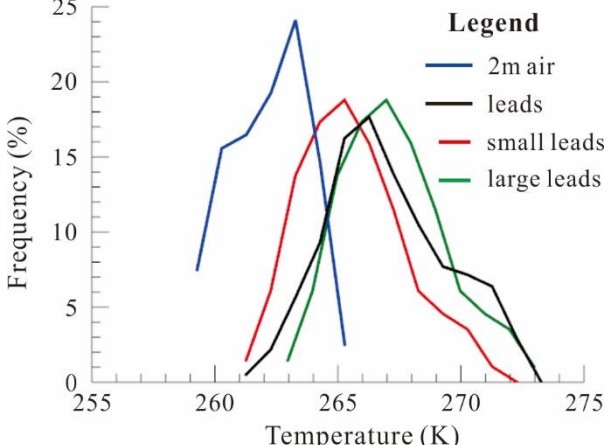

**Figure 12.** Distribution of 2 m air temperature over leads and surface temperature of all leads; small leads with width <1 km, larger leads with width <5 km.

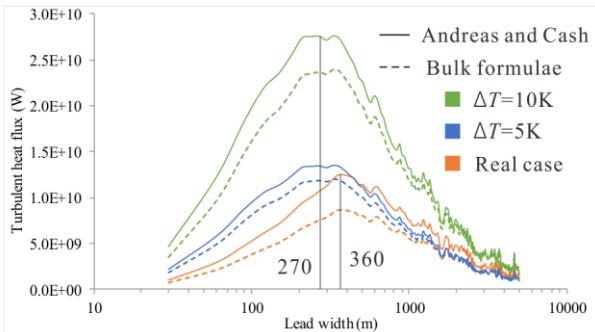

**Figure 13.** Contribution of heat flux from each lead width using bulk formulae and fetch-limited model. Turbulent heat flux retrieved using fetch-limited model and bulk formulae are plotted using solid and dashed lines, respectively. Heat flux calculated using satellite surface temperature, air temperature, and wind speed from reanalysis datasets is drawn in orange; simulated heat flux at $\Delta T$ = 5 K and 10 K is in blue and green, respectively.

**Tables**

**Table 1.** Satellite images and other data used in this study.

| Resource | Parameters | Spatial resolution | Time | Notes |
|---|---|---|---|---|
| Landsat-8 TIRS | Band 5 | 30 m | 21:27 | Near-infrared |
| | Band 8 | 15 m | 21:27 | Panchromatic |
| | Band 10 | 30 m | 21:27 | 10.60 μm – 11.19 μm |
| | Band 11 | 30 m | 21:27 | 11.50 μm – 12.51 μm |
| Terra MODIS | Band 31 | 1000 m | 20:55 | 10.78 μm – 11.28 μm |
| | Band 32 | 1000 m | 20:55 | 11.77 μm – 12.27 μm |
| ERA-interim Reanalysis | 10 m wind | 0.125°(~10 km) | 21:00 | 4.8~9.5 m/s |
| | 2 m air temperature | 0.125° (~10 km) | 21:00 | 259.3~265.6 K |
| | 2 m dew point Temperature | 0.125° (~10 km) | 21:00 | 257.3~263.8 K |

**Table 2.** Retrieved leads from MODIS and TIRS, and turbulent heat flux estimated using bulk formulae.

| Sensor | Lead category | Length (km) | Lead area | | Bulk formulae | |
|---|---|---|---|---|---|---|
| | | | (km²) | Contribution (%) | Heat flux (W) | Contribution (%) |
| MODIS | ≤1 km | 1050.0 | 1050.0 | 13.00 | 3.10E+10 | 5.79 |
| | 1 km~5 km | 1438.1 | 4065.0 | 50.35 | 1.97E+11 | 36.79 |
| | >5 km | 258.3 | 2959.0 | 36.65 | 3.08E+11 | 57.42 |

|  |  | Total | 2746.4 | 8074 |  | 5.36E+11 |  |
| --- | --- | --- | --- | --- | --- | --- | --- |
|  | TIRS | ≤1 km | 8502.2 | 2547.7 | 34.54 | 2.16E+11 | 25.75 |
|  |  | 1 km~5 km | 1440.7 | 2825.3 | 38.30 | 3.37E+11 | 40.09 |
|  |  | >5 km | 207.4 | 2003.3 | 27.16 | 2.87E+11 | 34.17 |
|  |  | Total | 10150.3 | 7376.2 |  | 8.40E+11 |  |

**Table 3.** Estimated turbulent heat flux (W) for Landsat-8 TIRS using bulk formulae, the Andreas and Cash (1999) model, and modified Andreas and Cash model using Obukhov length from Eq. (B8) and (B13) .

| Lead category | Bulk formulae | | Andreas and Cash (1999) | | Modified Andreas and Cash model | |
| --- | --- | --- | --- | --- | --- | --- |
|  | Heat flux | Contribution (%) | Heat flux | Contribution (%) | Heat flux | Contribution (%) |
| ≤1 km | 2.16E+11 | 25.75 | 3.06E+11 | 27.50 | 2.72E+11 | 27.99 |
| 1 km~5 km | 3.37E+11 | 40.09 | 4.43E+11 | 39.81 | 3.86E+11 | 39.75 |
| >5 km | 2.87E+11 | 34.17 | 3.63E+11 | 32.68 | 3.13E+11 | 32.25 |
| Total | 8.40E+11 |  | 1.11E+12 |  | 9.71E+11 |  |

**Table 4.** Threshold candidates for lead detection and corresponding lead fractions.

|  |  | Fixed1 | Fixed2 | Fixed3 | 1st Std | 2nd Std | 3rd Std | Iterative |
| --- | --- | --- | --- | --- | --- | --- | --- | --- |
| MODIS | Threshold (K) | 1 | 2 | 3 | 1.29 | 2.47 | 3.65 | 1.52 |
|  | Lead fraction (%) | 12.59 | 6.04 | 3.69 | 9.73 | 4.73 | 2.71 | 8.25 |
| TIRS | Threshold (K) | 1 | 2 | 3 | 1.90 | 3.52 | 5.14 | 2.49 |
|  | Lead fraction (%) | 14.85 | 8.65 | 6.62 | 8.93 | 5.69 | 2.82 | 7.53 |

**Table 5.** Turbulent heat flux (W) for higher temperature difference using Landsat-8 TIRS data and Andreas and Cash (1999) model.

|  | Lead category | Real case | | $\Delta T = 5$ K, $u_r = 7$ m/s | | $\Delta T = 10$ K, $u_r = 7$ m/s | |
| --- | --- | --- | --- | --- | --- | --- | --- |
|  |  | Heat flux | Contribution (%) | Heat flux | Contribution (%) | Heat flux | Contribution (%) |
| Bulk formulae | ≤1 km | 2.16E+11 | 25.75 | 2.92E+11 | 34.54 | 5.82E+11 | 34.54 |
|  | 1~5 km | 3.37E+11 | 40.09 | 3.24E+11 | 38.30 | 6.45E+11 | 38.30 |
|  | <5 km | 2.87E+11 | 34.17 | 2.30E+11 | 27.16 | 4.58E+11 | 27.16 |
|  | Total | 8.40E+11 |  | 8.46E+11 |  | 1.68E+12 |  |
| Andreas and Cash (1999) | ≤1 km | 3.06E+11 | 27.50 | 3.27E+11 | 35.86 | 6.66E+11 | 36.57 |
|  | 1~5 km | 4.43E+11 | 39.81 | 3.45E+11 | 37.88 | 6.85E+11 | 37.63 |
|  | <5 km | 3.63E+11 | 32.68 | 2.39E+11 | 26.25 | 4.69E+11 | 25.79 |
|  | Total | 1.11E+12 |  | 9.11E+11 |  | 1.82E+12 |  |

5   **Table A1.** Leads and pack ice pixels detected by Landsat-8 TIRS and panchromatic images at 15 m resolution.

| Panchromatic \ TIRS | Leads | Pack ice | Total | Producer's accuracy (%) |
| --- | --- | --- | --- | --- |
| Open water and thin ice | 27,039,061 | 3,172,911 | 30,211,972 | 89.5 |
| Refrozen lead | 4,710,542 | 41,620,953 | 46,331,495 |  |
| Pack ice | 471,960 | 368,561,891 | 369,033,851 |  |
| Total | 32,221,563 | 413,355,756 | 445,577,319 |  |
| User's accuracy (%) | 83.9 |  |  |  |