# Peer review of "Estimation of turbulent heat flux over leads using satellite thermal images"

_The Cryosphere, 2018_

## Referee Comment (RC1) · Anonymous Referee #1 · 28 Feb 2019

The authors present a study build upon widely used spaceborne thermal-infrared data from MODIS and Landsat-8 in combination with ECMWF ERA-Interim atmospheric reanalysis data to calculate turbulent heat fluxes. Based upon an almost perfectly co-located case study between the two sensors in the Beaufort Sea, the authors present a thorough analysis of the sensors capabilities for the detection of lead sizes and widths as well as a comparison between two different methodologies to calculate the turbulent heat fluxes. Overall, the manuscript is mostly well written and a good extension to existing work in the field.

General Comments:

Did the authors do anything about potentially present cloud cover? It looks to me that at least in some areas it could likely be a cloud artifact we are looking at.

[Figure]

Specific Comments

P1, L20: Does 'mainly due to its large area' refer to the area of small leads? Is that linked to a likelihood of rather being ice free than bigger leads?

P2, L44-45: Are Landsat-8 thermal bands really referred to as the 'split-window' bands?

P3, L15: Could the authors elaborate on their decision to not use the NSIDC MOD29 sea-ice surface temperature product directly but instead calculate it themselves using their parameters? Was it due to the applied cloud mask?

P6, L17: Could the authors discuss where this difference might originate from? From what I read this might simply be the difference between an optimized for sea-ice temperature scheme in comparison to a multi-purpose one?

P6, L21: I think I missed how exactly these iterative thresholds were calculated or estimated in the first place? In way to match the resulting lead sizes/distributions between the sensors? Iteratively implies for me that there is some kind of number/goal to reach.

P6, L25-26: Is this difference or rather the larger number for MODIS really simply just due to mixed pixels? Later on the authors discuss frequently how much of the total area comes from small leads, which MODIS cannot detect at all. From reading the manuscript, I would rather expect it to be different as there should not be any leads in MODIS that Landsat-8 cannot detect, but surely as the authors also stated, the other way around. Could clouds be a factor here?

P6, L30: Is the choice of lead-width thresholds arbitrary or is there a reference for that from another study?

P8, L16: Iterative thresholds are mentioned again but I think I still have not read an explanation yet.

P9, L19-20: Technically, MODIS cannot detect any leads in thermal infrared with a width below 1km? You compare numbers from below 1km width with numbers from

exactly 1km. I think that should be highlighted better or rephrased.

Technical Corrections

P1, L11: I think that should be 'scales'

P1, L20: 'flux over leads'

P1, L23: 'exposed to the atmosphere'

P7, L6-7: I suggest to rephrase this sentence(s): Table 2 reveals that the total heat flux over leadsn calculated using TIRS IST is 6.59[...] over the total area of [...]km$^2$. This is 42.33% larger [...]

P7, L14: Suggest to use 'difference' instead of 'increase'.

P7, L18/19: 'leads' and 'widths'. To my understanding, there are probably quite some more cases of that throughout the manuscript. The authors should double-check that.

P7, L33-35: I find this last sentence hard to comprehend. Please rephrase.

P7, L38: 'twice as' large?

P8, L8: 'to extract lead signatures from the background'

P8, L24: Second Key reference is not capitalized.

---

## Referee Comment (RC2) · Anonymous Referee #2 · 4 Mar 2019

A review on "Estimation of turbulent heat flux over leads using satellite thermal images"

The focus of the paper is the estimation of turbulent sensible and latent heat fluxes over leads using high-resolution satellite thermal images. The heat transfer over leads play important role in the heat budget of the atmospheric and oceanic boundary layers and affects many processes in the Arctic climate system. However, there is a large uncertainty in the estimates of turbulent heat flux over leads due several reasons: i) the insufficient resolution of satellite images used in models, ii) sparseness of in situ observations and iii) uncertainties in parametrizations of turbulent heat transfer over inhomogeneous sea ice surface. The paper provides new estimates of such uncertainties using satellite images of various resolution and shows the necessity to use high-resolution images and also more adequate parametrizations. To some extent, the

paper follows the line of the Marcq and Weiss (2012) paper, but uses realistic surface and air temperatures, as well as wind speed for their case study and also using different satellite data. Therefore, the study adds to the current knowledge and provides revised estimates of the heat flux calculation uncertainties and thus is relevant and valuable. However, the quality of the paper is low and has to be strongly improved. It concerns the choice and description of methods, the analysis of results and language. The paper cannot be accepted in its current state. I suggest major revision with resubmission.

Major comments

1. The two methods are used for the turbulent heat flux estimates: the traditional bulk formulae and the fetch-dependent model proposed by Andreas and Cash (1999). The bulk formulae and their application have to be described in more detail. First of all, it is potential temperature that has to be used in the formula for the sensible heat flux. Second, the heat transfer coefficients depend on height, surface roughness lengths for momentum and heat ($z_{0m}$ and $z_{0t}$) and stability. Which height, which values for the roughness lengths and, finally, which universal stability functions are used? The authors say that they use the air temperature at 2m height, but wind speed at 10m height from the reanalysis data. Since these heights differ from each other, the bulk formulae cannot be used in their classical form. The authors need to describe in detail how they solve the bulk equations. Do they use $z/L$ or the bulk Richardson number as a stratification parameter in the stability functions?

2. Concerning the fetch-dependent model. In lines 5-10 at page 5, the authors claim that the heat transfer over large leads is less efficient because the temperature (and humidity) difference between the lead surface and air is decreasing with fetch. This mechanism is present in the Renfrew and King (2000) model for heat fluxes over polynya (Renfrew, I.A. & King, J.C. Boundary-Layer Meteorology (2000) 94: 335. https://doi.org/10.1023/A:1002492412097) and in the model of Chechin and Lüpkes for cold-air outbreaks over the marginal sea-ice zone (Chechin, D. and Lüpkes, C. (2017), Boundary-Layer Meteorology, 162:91-116 , pp. 1-26 . doi: 10.1007/s10546-016-0193-

**TCD**

2 ). The authors should refer to these papers. However, in the basis of the Andreas and Cash model there is a different physical mechanism of how fetch affects turbulent heat transfer. Andreas and Cash suggest that the thicker the thermal boundary layer is, the closer the conditions are to the free-convective limit. They claim that in the free-convective limit the heat transfer is less efficient than in the forced convection. I suggest, that the authors review the existing physical interpretations of the effect of fetch, e.g. by Andreas and Cash, by Alam and Curry (1997), which are different. Also (!), in the Andreas and Murphy (1986) paper, different physics is described (e.g., the effect of a more rough sea ice, for example, and a different interpretation of the free-convection contribution). Also, refer to the Esau 2007 paper (Amplification of turbulent exchange over wide Arctic leads: Large‐eddy simulation study, J. Geophys. Res., 112, D08109, doi:10.1029/2006JD007225. )

3. As already mentioned, there is another model which takes into account the dependency of heat flux on fetch over leads, namely, the Alam and Curry (1997) model. This model has different physics and more processes are taken into account. It is not clear why the authors prefer the Andreas and Cash model and do not consider the Alam and Curry model. This has to be explained.

4. One of the results of the study is that the fetch-dependent model produces larger fluxes than the bulk formulae. However, the transfer coefficients in the bulk formulae depend strongly on the roughness length for momentum and heat ($z_{0m}$ and $z_{0t}$) and therefore, the obtained result is only valid for specific values of $z_{0m}$ and $z_{0t}$, which are not given in the paper(!). Using other values for $z_{0m}$ and $z_{0t}$ can produce completely different results. The Andreas and Cash model, as it is described in the paper, does not show an explicit dependency on the roughness length. However, implicitly, roughness is present in their model and the authors need to describe how the roughness length is present in the model of Andreas and Cash. What are the values for the roughness length in the Andreas and Cash model and how do they compare with the ones used in the bulk formulae? Note, that the Andreas and Cash model is a refor-

mulation of the earlier Andreas and Murphy model. The latter is formulated in such a way that it is compatible with bulk formulae. Namely, the are suggesting to use a fetch-dependent "lead-averaged" neutral heat transfer coefficient. In other words, the Andreas and Murphy formulation would allow a more reasonable comparison with the standard bulk approach.

5. Describe better the case study. Which date is it, what are the synoptic conditions over the study area? Was it a clear-sky case or clouds were present? Does it represent typical conditions in the Arctic? The presented surface and air temperature distributions suggest that this is either autumn or late spring. But one would expect that the effect of leads is strongest in winter.

Minor comments

–Page 1, lines 26-27, rephrase "The rate of turbulent heat transferred", simply "Turbulent heat flux"

–Page 1, line 36. "More intensive network" needs clarification. Also, "stronger influence of leads" - influence on what?

– Page 2, line 14 - "heat flux transfer rate" - the efficiency of heat transfer?

– Page 2 line 16 – remove "More often than not"

– Page 2, lines 20-25, explain better what is meant by "Fetch limited models" and how they are using the fetch-dependence of the internal boundary layer height. Otherwise, the logic is disrupted.

– At least in the introduction the authors should cite the study by Tetzlaff et al. (2015) where the most recent observations of heat fluxes and the internal boundary layer height over leads are presented: Tetzlaff, A. , Lüpkes, C. and Hartmann, J. (2015), Aircraftâ Řbased observations of atmospheric boundaryâ Řlayer modification over Arctic leads. Q.J.R. Meteorol. Soc., 141: 2839-2856. doi:10.1002/qj.2568

– Page 3, line 10. The actual grid of the ERA-Interim reanalysis has horizontal spacing 0.75o and not 0.125o. The original ERA Interim data is interpolated on the 0.125o grid which does not increase the resolution.

– Page 3, line 39. albedo anomaly

– Page 4, Lines 8-9 rephrase "varying air condition"

– Page 4, Line 21. "limited used of lead width" - what does it mean??

– Page 4, Line 29 "rate of turbulent heat change" - what does it mean?? Rephrase!

– Page 4, Eq. 4 – use CH instead of Cs for the heat transfer coefficient

– Page 4, Line 36 Ce is the bulk transfer coefficient for water vapor

– Page 4, Line 37 "at the surface", this is wrong. These values are at heights z0t and z0q, which are the roughness lengths for heat and moisture fluxes.

– Page 4, Do the authors use the saturation specific humidity over ice or over water? Do they distinguish between the open water and thin ice in this respect?

– Page 5, Line 11. This is shown in figure 4, not in figure 3.

– Page 5, Line 35, Not "in spite", but "apart from a dependency on the width of a lead"

– Page 5, line 36. Which simulation? What was used for this simulation - Bulk formulae, or the Andreas and Cash model? For the bulk model, such result is obvious and does not need to be shown. Andreas and Cash write that their model is, on the contrary, not very sensitive to wind speed.

– Page 6, Section name "Results" and not "Result".

– Page 6, line 35. How is the length of leads calculated?

– Page 6, line 35. The Authors say that the MODIS resolution is 1000m. However, they introduce a class of small (which they call narrow in other places) leads with width less

or equal 1km. Somehow, they found 13% of such leads in the MODIS image. They need to comment if 1km wide leads are resolved with 1km resolution.

– Page 7, line 4. The direction of fluxes from the ocean to the atmosphere is not consistent with the Eq. (4) and (5). The authors should modify the Equations to get the right sign and direction of fluxes.

– Page 7 line 12. In which range of width is the fetch-limited model valid? Why was it applied to Landsat only? There wide leads in Landsat as well. Write that Landsat better resolves leads, so that's why it was applied to it.

---

## Author Comment (AC2) · 19 Apr 2019

The authors would very much like to thank you for your time and constructive comments. We have considered each comment carefully and incorporated practically all of them. Please find attached a revised version of our manuscript, in which we marked major modifications in red. Response to each comment are as follows:

"A review on "Estimation of turbulent heat flux over leads using satellite thermal images" The focus of the paper is the estimation of turbulent sensible and latent heat fluxes over leads using high-resolution satellite thermal images. The heat transfer over leads play important role in the heat budget of the atmospheric and oceanic boundary layers and affects many processes in the Arctic climate system. However, there is a

[Figure]

large uncertainty in the estimates of turbulent heat flux over leads due several reasons: i) the insufficient resolution of satellite images used in models, ii) sparseness of in situ observations and iii) uncertainties in parametrizations of turbulent heat transfer over inhomogeneous sea ice surface. The paper provides new estimates of such uncertainties using satellite images of various resolution and shows the necessity to use high-resolution images and also more adequate parametrizations. To some extent, the paper follows the line of the Marcq and Weiss (2012) paper, but uses realistic surface and air temperatures, as well as wind speed for their case study and also using different satellite data. Therefore, the study adds to the current knowledge and provides revised estimates of the heat flux calculation uncertainties and thus is relevant and valuable. However, the quality of the paper is low and has to be strongly improved. It concerns the choice and description of methods, the analysis of results and language. The paper cannot be accepted in its current state. I suggest major revision with resubmission."

"Major comments"

"1.The two methods are used for the turbulent heat flux estimates: the traditional bulk formulae and the fetch-dependent model proposed by Andreas and Cash (1999).

The bulk formulae and their application have to be described in more detail.

First of all, it is potential temperature that has to be used in the formula for the sensible heat flux.

Second, the heat transfer coefficients depend on height, surface roughness lengths for momentum and heat (z0m and z0t) and stability. Which height, which values for the roughness lengths and, finally, which universal stability functions are used?

The authors say that they use the air temperature at 2m height, but wind speed at 10m height from the reanalysis data. Since these heights differ from each other, the bulk formulae cannot be used in their classical form.

The authors need to describe in detail how they solve the bulk equations. Do they

use z/L or the bulk Richardson number as a stratification parameter in the stability functions?"

Reply: Thanks for your questions and advices. As suggested, the bulk formulae are now modified in Page 5, Line13 ∼ 17.

Page 5, Line 24 ∼ 25, we added: "Csh and Cle are transfer coefficients for sensible heat and latent heat, calculated using equations from Oberhuber (1988) and Goosse et al. (2000) (see Appendix B)."

In our study, 2m air temperature from ERA-interim Reanalysis datasets was used as potential temperature Tr in the formulae;

A constant turbulent heat coefficient of $1.44 \times 10^{(-3)}$ from Nihashi and Ohshima (2001) was used in our previous experiment, which might not be appropriated for Arctic leads. As suggested, we have modified the experiments using equations (Appendix B) from Oberhuber (1988), Goose et al. (2000), and Marcq and Weiss (2012) to solve the coefficients for the bulk formulae. Most part of the parameterization originates from Large and Pond (1981, 1982):

Page 11, Line 2 ∼ 18, we added Appendix B:"Equations used for turbulent heat flux estimation using bulk formulae (Large and Pond, 1981, 1982; Oberhuber, 1988; Goosse et al, 2000; Marcq and Weiss, 2012)"

Since the wind speed and air temperature from ERA-interim are at different height, in our previous manuscript, a power law equation was used to calculate ur using 10m wind magnitude u10 from ERA-interim:

$u_r/u_{10} = (r/10)^a$

where a is the wind shear exponent. An empirical value of a = 1/7 was used. In our modified experiments, a log law equation based on principles of boundary layer flow, was used (Ray et al., 2006):

Page 5, line 26∼29, we added: Since the wind speed and air temperature from ERA-interim are at different heights, a wind profile equation was used (Ray et al., 2006):

$u10/ur = (\ln 10 - \ln Z0)/(\ln r - \ln Z0)$ (9)

where $u10$ and $ur$ are wind speed at 10m and 2m height, and $Z0$ is the surface roughness length."

As a result, the mean wind speed at 2m height over leads, rises from ∼5m/s (using power law) to ∼7m/s (using log law). Estimated turbulent heat flux also increases in response.

Assuming an initial value for friction velocity $u^*$, the equations (B1) ∼ (B13) and Eq. (9) are solve iteratively. As shown in Eq. (B13), the equations use $z/L$ as a stratification parameter, comparing to the bulk Richardson number used in Andreas and Cash (1999).

Reference:

Nihashi S, Ohshima K I.: Relationship between ice decay and solar heating through open water in the Antarctic sea ice zone. Journal of Geophysical Research: Oceans, 106: 16767-16782. 2001.

Oberhuber, J. M.: An atlas based on the 'Coads' data set: The budgets of heat, buoyancy and turbulent kinetic energy at the surface of global ocean. 1988.

Goosse, H., Campin, J. M., Deleersnijder, E., Fichefet, T., Mathieu, P. P., Maqueda, M. M., and Tartinville, B.: Description of the CLIO model version 3.0. Institut d'Astronomie et de Géophysique Georges Lemaitre, Catholic University of Louvain, Belgium. 2001.

Marcq, S., and Weiss, J.: Influence of sea ice lead-width distribution on turbulent heat transfer between the ocean and the atmosphere. The Cryosphere, 6, 143-156, 2012.

Large, W. G., and Pond, S.: Open ocean momentum flux measurements in moderate to strong winds. Journal of physical oceanography, 11(3), 324-336. 1981.

Large, W. G., and Pond, S.: Sensible and latent heat flux measurements over the ocean. Journal of physical Oceanography, 12(5), 464-482. 1982.

Ray, M. L., Rogers, A. L., and McGowan, J. G.: Analysis of wind shear models and trends in different terrains. University of Massachusetts, Department of Mechanical and Industrial Engineering, Renewable Energy Research Laboratory. 2006.

"2. Concerning the fetch-dependent model. In lines 5-10 at page 5, the authors claim that the heat transfer over large leads is less efficient because the temperature (and humidity) difference between the lead surface and air is decreasing with fetch. This mechanism is present in the Renfrew and King (2000) model for heat fluxes over polynya (Renfrew, I.A. & King, J.C. Boundary-Layer Meteorology (2000) 94: 335. https://doi.org/10.1023/A:1002492412097) and in the model of Chechin and Lüpkes for cold-air outbreaks over the marginal sea-ice zone (Chechin, D. and Lüpkes, C. (2017), Boundary-Layer Meteorology, 162:91-116 , pp. 1-26 . doi: 10.1007/s10546-016-0193-2). The authors should refer to these papers. However, in the basis of the Andreas and Cash model there is a different physical mechanism of how fetch affects turbulent heat transfer. Andreas and Cash suggest that the thicker the thermal boundary layer is, the closer the conditions are to the free-convective limit. They claim that in the free-convective limit the heat transfer is less efficient than in the forced convection. I suggest, that the authors review the existing physical interpretations of the effect of fetch, e.g. by Andreas and Cash, by Alam and Curry (1997), which are different. Also (!), in the Andreas and Murphy (1986) paper, different physics is described (e.g., the effect of a more rough sea ice, for example, and a different interpretation of the free convection contribution). Also, refer to the Esau 2007 paper (Amplification of turbulent exchange over wide Arctic leads: Large eddy simulation study, J. Geophys. Res., 112, D08109, doi:10.1029/2006JD007225. )"

Reply: Thanks for these valuable suggestions, a brief review on recommended studies was added to our manuscript.

Page 2, Line27 ∼ 40, we added: "Models were developed for estimation of TIBL thickness and turbulent heat flux over coastal polynyas, leads, and ice edges (Alam and Curry, 1997; Andreas and Cash, 1999; Renfrew and King, 2000, Chechin and Lüpkes, 2017). Chechin and Lüpkes (2017) modeled boundary layer development downwind of the ice edge, potential temperature, and mix-layer height, and wind speed variation was analyzed as well. Renfrew and King (2000) modeled turbulent heat flux over large fetch (5–50 km wide, typical for coastal polynya) during cold-air outbreaks. The dependence of turbulent heat flux on lead width was estimated in several studies (Andreas and Murphy, 1986; Alam and Curry, 1997; Andreas and Cash, 1999). On the basis of the Monin–Obukhov similarity theory and the surface renewal theory, Alam and Curry (1997) estimated turbulent heat flux over leads using an intricate surface roughness model (Bourassa et al., 2001). Sensible heat flux across a single lead is integrated from fetch 0 to fetch X. Andreas and Murphy (1986) calculated transfer coefficient CN10 at 10 m height for turbulent heat in neutral stability, using the nondimensional fetch –X/L, where L is the Obukhov length. A maximum CN10 of $1.8 \times 10^{(-3)}$ was found at small fetch, and the value decrease to $1.0 \times 10^{(-3)}$ with increasing –X/L. Andreas and Cash (1999) computed lead-average turbulent heat flux using transfer coefficient C* as a function of stability parameter –h/L, where h is the fetch-dependent height of the TIBL. For small fetch (–h/L < 6), turbulent heat is exchanged by mixed free and forced convection, resulting in a large C* and higher heat flux."

Page 5, Line38 ∼ 40, we added: "Another mechanism is described in Esau (2007) for leads 1km∼10km wide. Under weak wind condition (∼2 m/s), convective overturning prevents cold breezes from penetrating into the lead area, reducing the average turbulent heat flux."

"3. As already mentioned, there is another model which takes into account the dependency of heat flux on fetch over leads, namely, the Alam and Curry (1997) model. This model has different physics and more processes are taken into account. It is not clear why the authors prefer the Andreas and Cash model and do not consider the Alam and

Curry model. This has to be explained."

Reply: Thanks for pointing out the Alam and Curry (1997) model. In that model, turbulent heat flux across single lead is integrated from fetch 0 to fetch X, along the wind direction. Indeed, it is more theoretical and takes more factors into account. But in the large scale application, it is hardly applicable due to lack of high resolution meteorological data like 2m air temperature. Besides, the model assumes universal water surface within leads (with a complicated surface roughness model for open ocean), which is different from our case where narrow open water or thin ice dominate. Actually, we tried to apply the Alam and Curry (1997) model in the remote sensing setting, but failed due to lack of sea surface information (e.g. phase speed, wave age etc.). Thus, only the fetch-limited model of Andreas and Cash (1999) was selected in our experiment.

Page 6, Line 2 $\sim$ 8, we added the explanations as follows: "However, the assumption of universal water surface in leads and the application of sea surface roughness model (Andreas and Murphy, 1986; Alam and Curry, 1997) are not applicable in our case, where open water and thin ice dominate. Since the signal of TIBL is absent in the coarse grid of 2 m air temperature from the ERA reanalysis dataset, the data might not be appropriate to demonstrate the Alam and Curry (1997) model, which relies on accurate measurement of meteorological parameters. Whereas the Andreas and Cash (1999) model is more sensitive to lead width than atmospheric conditions (Marcq and Weiss, 2012). Therefore, only the Andreas and Cash (1999) model was used in our experiment."

"4. One of the results of the study is that the fetch-dependent model produces larger fluxes than the bulk formulae. However, the transfer coefficients in the bulk formulae depend strongly on the roughness length for momentum and heat ($z0m$ and $z0t$) and therefore, the obtained result is only valid for specific values of $z0m$ and $z0t$, which are not given in the paper (!). Using other values for $z0m$ and $z0t$ can produce completely different results. The Andreas and Cash model, as it is described in the paper, does not show an explicit dependency on the roughness length. However, implicitly,
roughness is present in their model and the authors need to describe how the roughness length is present in the model of Andreas and Cash. What are the values for the roughness length in the Andreas and Cash model and how do they compare with the ones used in the bulk formulae? Note, that the Andreas and Cash model is a reformulation of the earlier Andreas and Murphy model. The latter is formulated in such a way that it is compatible with bulk formulae. Namely, they are suggesting to use a fetch-dependent "lead-averaged" neutral heat transfer coefficient. In other words, the Andreas and Murphy formulation would allow a more reasonable comparison with the standard bulk approach."

Reply: Thanks for the question. For open water leads, Andreas and Murphy (1986) parameterize turbulent transfer coefficient at 10m as

CN10 = (1.0+0.8 EXP (0.05(X/L)))/1000

where X is the fetch or lead width. However, the reference height of 10m might not be suitable for study narrow leads in pack ice, where the TIBL is generally shallower. On the base of Andreas and Murphy (1986), Andreas and Cash (1999) used bulk Richardson number Rib to calculate Obukhov Length L i.e. Eq. (16) under the assumption that the ratio between roughness lengths for momentum and heat, i.e. Z0 / ZT, is about EXP(2). In our case, this simplification result in a higher |L| than that from Oberhuber (1988) and Goosse et al. (2000). If the Obukhov length from Eq. (B8) and (B13) were used to calculate the coefficient C* in Andreas and Cash (1999) model, and estimated turbulent heat flux will be smaller (Table 3), but still 15.53% larger than that from bulk formulae.

Page 9 line 25∼31, we added: "In both the Andreas and Murphy (1986) and Andreas and Cash (1999) models, for reference height r < 10 m, the ratio between roughness lengths for momentum and heat, Z0/ZT, is assumed to be ∼EXP(2) to calculate Obukhov length L using Richardson number Rib (see Eq. (17)). The calculated Obukhov length L has absolute values about 67% higher than those using Eq. (B8) and

(B13) from the bulk formulae (Oberhuber, 1988; Goosse et al., 2000). If Eq. (B8 ) and (B13) were used to solve Obukhov length and coefficient C* in the Andreas and Cash (1999) model, estimated turbulent heat flux will be smaller (Table 3), but still 15.53% larger than that from the bulk formulae, with an even larger part of the difference from the small lead category (42.48%, compared to 32.96% in Section 4.3.2)."

Page 6, Line 27, we have Eq. (17)

$L^{-1} = 8.0*(0.65/r+0.079-0.0043r)*Rib$ (17)

Page 11, Line 11 and 18, we have Eq. (B8) and (B13)

$T0 = Tr \times (1+2.2 \times Tr \times qr \times 10^{-3})$ (B8)

$r/L = 100r \times ((Ts-Tr)+2.2 \times (T0^2)(qs-qr) \times 10^{-3})/(T0 \times ur^2)$ (B13)

Page 21, we updated Table 3: Estimated turbulent heat flux (W) for Landsat-8 TIRS using bulk formulae, the Andreas and Cash (1999) model, and modified Andreas and Cash model using Obukhov length from Eq. (B8) and (B13).

"5. Describe better the case study. Which date is it, what are the synoptic conditions over the study area? Was it a clear-sky case or clouds were present? Does it represent typical conditions in the Arctic? The presented surface and air temperature distributions suggest that this is either autumn or late spring. But one would expect that the effect of leads is strongest in winter."

Reply: Thanks for the question. The Landsat-8 and MODIS images used in our study were acquired on April 26, 2016. The study area is mostly unobstructed by cloud in this scene. According to the ERA-interim datasets, the study area is dominated by polar easterlies, with 10m wind speed range from 4.8 to 9.5 m/s. Air temperatures at 2m from the reanalysis data range from 257.3 $\sim$ 263.8K. Temperature difference between surface and 2m air is about 5K. The Landsat-8 imagery is sparse in the Arctic ocean, especially in the winter polar night. These three successive scenes of thermal images can also provide valuable details of spatial distribution of spring leads.

Page 3, Line 6, we added: "acquired on April 26, 2016"

Page 20, we updated table 1:"Satellite images and other data used in this study."

"Minor comments"

–Page 1, lines 26-27, rephrase "The rate of turbulent heat transferred", simply "Turbulent heat flux"

Corrected

–Page 1, line 36. "More intensive network" needs clarification. Also, "stronger influence of leads" - influence on what?

Corrected, "networks of more intensive lead with stronger local influence are expected"

– Page 2, line 14 - "heat flux transfer rate" - the efficiency of heat transfer?

Corrected, "Assuming higher heat transfer over narrow leads than wider leads"

– Page 2 line 16 – remove "More often than not"

Corrected

– Page 2, lines 20-25, explain better what is meant by "Fetch limited models" and how they are using the fetch-dependence of the internal boundary layer height. Otherwise, the logic is disrupted.

A review of models developed for estimation of internal boundary layer height and turbulent heat flux was added on page 2 Line $27 \sim 40$.

– At least in the introduction the authors should cite the study by Tetzlaff et al. (2015) where the most recent observations of heat fluxes and the internal boundary layer height over leads are presented: Tetzlaff, A. , Lüpkes, C. and Hartmann, J. (2015), Aircraft based observations of atmospheric boundary layer modification over Arctic leads. Q.J.R. Meteorol. Soc., 141: 2839-2856. doi:10.1002/qj.2568

Page2, Line 25 $\sim$ 26, we added: "Convective plumes formed above leads may further complicate the process within the TIBL (Tetzlaff et al., 2015)."

– Page 3, line 10. The actual grid of the ERA-Interim reanalysis has horizontal spacing 0.75o and not 0.125o. The original ERA Interim data is interpolated on the 0.125o grid which does not increase the resolution.

Corrected Page3, Line 9, we write: "This dataset provides global coverage with a temporal resolution of 3 hours, 0.125° grid data is available for download ($\sim$10 km in study area, interpolated from original 0.75° grid)."

– Page 3, line 39. albedo anomaly

Corrected

– Page 4, Lines 8-9 rephrase "varying air condition"

Corrected, "air temperature variation"

– Page 4, Line 21. "limited used of lead width" - what does it mean??

Deleted

– Page 4, Line 29 "rate of turbulent heat change" - what does it mean?? Rephrase!

Corrected, "turbulent heat exchange"

– Page 4, Eq. 4 – use CH instead of Cs for the heat transfer coefficient

Corrected

– Page 4, Line 36 Ce is the bulk transfer coefficient for water vapor

Corrected

– Page 4, Line 37 "at the surface", this is wrong. These values are at heights z0t and z0q, which are the roughness lengths for heat and moisture fluxes.

Corrected.

– Page 4, Do the authors use the saturation specific humidity over ice or over water? Do they distinguish between the open water and thin ice in this respect?

Yes. Saturated specific humidity was used over lead surface for both open water and thin ice, but different sets of parameter were used to calculate saturated vapor pressure at the surface of open water and thin ice.

Page 5, Line 20∼23, we write: "es0 represents the saturated vapor pressure at surface temperature Ts:

es0 = e0×10ˆ(at/(b+t)) (8)

with e0 represent saturate vapor pressure at 0 °C, approximately 6.11 hPa, t is temperature in Celsius, and a and b are coefficients (for water surface, a = 7.5, b = 237.3 K; for ice, a = 9.5, b = 265.5 K)."

– Page 5, Line 11. This is shown in figure 4, not in figure 3.

Corrected

– Page 5, Line 35, Not "in spite", but "apart from a dependency on the width of a lead"

Corrected

– Page 5, line 36. Which simulation? What was used for this simulation - Bulk formulae, or the Andreas and Cash model? For the bulk model, such result is obvious and does not need to be shown. Andreas and Cash write that their model is, on the contrary, not very sensitive to wind speed.

The simulation, as well as Fig. 3 and 4, is based on Andreas and Cash (1999) parameterization. Although, they claim that their model "depends only weakly on surface level wind speed". Our test shows that, for the narrowest lead from TIRS (X=30m), turbulent heat flux, especially sensible heat, rises quickly with larger âŰşT and stronger wind.

– Page 6, Section name "Results" and not "Result".

Corrected

– Page 6, line 35. How is the length of leads calculated?

Page 4, Line 43 ∼ page 5, Line 2 we added: Since we assign lead width to every pixel across the lead, the length Li for lead width Xi can be calculated as follow:

Li = a0×Ni/Xi (4)

where a0 is the pixel size, for TIRS, the value is 30 m, for MODIS, 1km; and Ni is pixel number for width Xi = a0×i, (i = 1, 2, 3...).

– Page 6, line 35. The Authors say that the MODIS resolution is 1000m. However, they introduce a class of small (which they call narrow in other places) leads with width less or equal 1km. Somehow, they found 13% of such leads in the MODIS image. They need to comment if 1km wide leads are resolved with 1km resolution.

Reply: The detectability of leads are composite of thresholds and contrast in surface temperature of leads compared to the surrounding ice, i.e. temperature anomaly ∆t. Although the 1km resolution is the finest for MODIS thermal (and AVHRR), potential and subpixel lead can be detected at this scale (Lindsay and Rothrock, 1995). In the revised version, we now include 1km in the first category, and statistics in Table 2 are updated.

Page7, Line 24 ∼ 26, we added: "Although, the 1km resolution is the finest for MODIS thermal, the 1km-wide lead category should provide a reasonable guess of potential small leads or subpixel leads at MODIS scale (Lindsay and Rothrock, 1995)."

Page 9, Line19 ∼ 23, we added: "In comparison with Landsat-8 TIRS and panchromatic images, we find that the lead map generated from the MODIS IST data neglects very small leads, but overestimates the width of other leads approximately 1 km wide. Overall, the 1 km wide lead category at MODIS scale should provide a reasonable

guess of potential small or subpixel leads. The small leads retrieved using TIRS provide a valuable reference for the capacity of MODIS to detect narrow leads."

Lindsay, R. W., and Rothrock, D. A.: Arctic sea ice leads from advanced very high resolution radiometer images. Journal of Geophysical Research: Oceans, 100(C3), 4533-4544, 1995.

– Page 7, line 4. The direction of fluxes from the ocean to the atmosphere is not consistent with the Eq. (4) and (5). The authors should modify the Equations to get the right sign and direction of fluxes.

Corrected.

– Page 7 line 12. In which range of width is the fetch-limited model valid? Why was it applied to Landsat only? There wide leads in Landsat as well. Write that Landsat better resolves leads, so that's why it was applied to it.

Reply: Andreas and Cash (1999) fit C* for -h/L range from 0.05 to 20. Although high wind ($\sim$7m/s) and low temperature difference ($\sim$5K) in our case leading to large |L| up to 40m, the C* fitting still holds for lead width X > 15m from TIRS. Since the turbulent heat flux saturate for lead width great than 1km, as depicted in Fig. 4, we think there is no need to apply the model with coarse leads from MODIS.

Please also note the supplement to this comment:
https://www.the-cryosphere-discuss.net/tc-2018-262/tc-2018-262-AC2-supplement.pdf

---

## Author Response (AR1)

In this documents, the authors provide response to Reviewers' Comments of paper ta-2018-262.

Qu, M. and Pang, X. and Zhao, X. and Zhang, J. and Ji, Q. and Fan, P.: Estimation of turbulent heat flux over leads using satellite thermal images, The Cryosphere Discussions., https://doi.org/10.5194/tc-2018-262, in review, 2019.

The authors would very much like to thank the reviewers for their time and constructive comments. We have considered each comment carefully and incorporated practically all of them. Responses to each comment and a marked-up manuscript are as follows:

**Reviewer 1:**

*"The authors present a study build upon widely used space borne thermal-infrared data from MODIS and Landsat-8 in combination with ECMWF ERA-Interim atmospheric reanalysis data to calculate turbulent heat fluxes. Based upon an almost perfectly collocated case study between the two sensors in the Beaufort Sea, the authors present a thorough analysis of the sensors capabilities for the detection of lead sizes and widths as well as a comparison between two different methodologies to calculate the turbulent heat fluxes. Overall, the manuscript is mostly well written and a good extension to existing work in the field."*

**General Comments:**

*"Did the authors do anything about potentially present cloud cover? It looks to me that at least in some areas it could likely be a cloud artifact we are looking at."*

Reply: Thanks for the question. For comparison, ice surface temperature (IST) map from MOD29 product is plotted in Fig.1. Potential cloud pixels are removed in MOD29 using cloud mask from MOD35 product, shown as "Nodata" in green color in Fig1 (b) and (C). As we can see, the pixels within leads marked as cloud are likely open water lead with fog or plume over the surface (Fett et al., 1997). To reserve potential lead areas, we applied the NSIDC algorithm (Hall et al. 2001) on thermal images from MODIS L1B product to calculate IST instead of using the MOD29 product. Since the area within Landsat-8 frame is mostly unobstructed by cloud, no cloud mask procedure was performed in our study.

[Figure]

Figure 1 Comparison of MODIS L1B thermal image (a) and MOD29 IST product (b), detail of Landsat-8 frame area in (b) are shown in (c).

In Page 3, Line 13 ~19, we added:

"Willmes and Heinamann (2015) used the MOD29 ice surface temperature (IST) product (Hall and Riggs, 2015) from the National Snow and Ice Data Center (NSIDC) to retrieve leads. The MOD29 product is filtered for cloud contamination using a cloud mask from MOD35. However, inspection of the corresponding MOD29 map of the study area revealed that the pixels within leads marked as clouds are likely open water with ocean fog or plume over the surface (Fett et al., 1997). Apart from that, the study area within the Landsat-8 frame is mostly unobstructed by clouds. To preserve potential lead areas, we applied the NSIDC algorithm (Hall et al. 2001) to thermal images from MODIS L1B to calculate IST instead of using the MOD29. Therefore, no cloud mask procedure was performed in our study."

Reference:
Fett, R. W., Englebretson, R. E., and Burk, S. D.: Techniques for analyzing lead condition in visible, infrared and microwave satellite imagery. Journal of Geophysical Research: Atmospheres, 102(D12), 13657-13671, 1997.
Hall, D. K. and G. Riggs.: MODIS/Terra Sea Ice Extent 5-Min L2 Swath 1km, Version 6. Boulder, Colorado USA. NASA National Snow and Ice Data Center Distributed Active Archive Center. doi: https://doi.org/10.5067/MODIS/MOD29.006. 2015.
Hall, D. K., Riggs, G. A., Salomonson, V. V., Barton, J. S., Casey, K., Chien, J. Y. L., ... and Tait, A. B.: Algorithm theoretical basis document (ATBD) for the MODIS snow and sea ice-mapping algorithms. Nasa Gsfc, 45, 2001.

**Specific Comments:**

*"P1, L20: Does 'mainly due to its large area' refer to the area of small leads? Is that linked to a likelihood of rather being ice free than bigger leads?"*
Reply: Yes, the phrase 'mainly due to its large area' refers to the total area of small leads. However, as explained in the manuscript, within any remote sensing pixel, the radiometric signature of a narrow lead with open water may be identical to that of a wider lead with thin ice. Since the surface temperature of narrow leads from Landsat-8 are mostly below the freezing point (see figure below), we are not sure whether the temperature signature is caused by subpixel open water lead or just lead covered with thin ice.

Page 8, line23, we write:
"However, within any optical, thermal or microwave image, the radiometric signature of a narrow lead with open water may be identical to that of a wider lead with thin ice."

Page 20, Line 5, we have Fig.12:

[Figure]

Figure 12. Distribution of 2 m air temperature over leads and surface temperature of all leads, small leads with width <1km, and larger leads with width >5km.

*"P2, L44-45: Are Landsat-8 thermal bands really referred to as the 'split-window' bands?"*

Reply: Yes. Although radiance measured by Landsat-8 Thermal Infrared Sensor (TIRS) suffers from stray light, it can observe ocean surface using two narrow thermal bands centered around 11μm and 12μm, more like the channel 4 & 5 on AVHRR and band 31 & 32 on MODIS rather than the high gain and low gain mode available for ETM+ channel 6 aboard Landsat-7. Such design is referred to as 'split-window' channels or bands in literatures about AVHRR and MODIS, so we consider to use it here.

*"P3, L15: Could the authors elaborate on their decision to not use the NSIDC MOD29 sea-ice surface temperature product directly but instead calculate it themselves using their parameters? Was it due to the applied cloud mask?"*

Reply: Yes, it is all about the cloud mask. As explained above, the NSIDC MOD29 IST product filtered for cloud contamination using cloud mask from MOD35 product, tends to mark open water leads as cloud in the presence of ocean fog or plume (see the Fig. 1 above), thus we avoid MOD29 for this special purpose.

In Page 3, Line 13 ~19, we added:

"Willmes and Heinamann (2015) used the MOD29 ice surface temperature (IST) product (Hall and Riggs, 2015) from the National Snow and Ice Data Center (NSIDC) to retrieve leads. The MOD29 product is filtered for cloud contamination using a cloud mask from MOD35. However, inspection of the corresponding MOD29 map of the study area revealed that the pixels within leads marked as clouds are likely open water with ocean fog or plume over the surface (Fett et al., 1997). Apart from that, the study area within the Landsat-8 frame is mostly unobstructed by clouds. To preserve potential lead areas, we applied the NSIDC algorithm (Hall et al. 2001) to thermal images from MODIS L1B to calculate IST instead of using the MOD29. Therefore, no cloud mask procedure was performed in our study."

*"P6, L17: Could the authors discuss where this difference might originate from? From what I read this might simply be the difference between an optimized for sea-ice temperature scheme in comparison to a multi-purpose one?"*

Reply: The difference in IST maps retrieved from MODIS and Landsat-8 TIRS, might result from, as suggested, the algorithms, the thermal radiance measured by the two different sensors, as well as any calibration procedure. As for the difference in algorithms, apart from their application range, we can find that sensor viewing angles are considered in Key's equation for IST retrieval from AVHRR and MODIS (mainly due to their wide swath), but not in the equation for TIRS. However, their difference has little impact on temperature anomaly and lead detection. A comprehensive comparison of IST retrieved from MODIS and TIRS is not the main goal of this study.

Page 8, Line 1 ~ 2, we added:

"About 23% of the difference (in turbulent heat flux estimated using bulk formulae) can be explained by IST bias between MODIS and TIRS, but most of the difference comes from small leads."

*"P6, L21: I think I missed how exactly these iterative thresholds were calculated or estimated in the first place? In way to match the resulting lead sizes/distributions between the sensors? Iteratively implies for me that there is some kind of number/goal to reach."*

Reply: Thanks for pointing this out. Several image-based threshold selection techniques for a binary lead segmentation were compared in Willmes and Heinemann (2015), and an iterative threshold selection method (Ridler and Calvard, 1978) was recommended for extracting leads from temperature anomaly map.

Assuming an initial threshold using the mean ($m_0$) of the whole image, the iterative threshold selection method proposed by Ridler and Calvard (1978) seeks for a threshold, $m_i$, which is the mean of two averages $m_A$ and $m_B$ from two clusters divided by $m_i$: target A and background B. In our study, this iterative process was performed in an IDL procedure.

Page 4, Line 33 ~ 34, we added the following explanations:

"Assuming an initial threshold using the mean temperature anomaly ($m_0$) of the whole image, the iterative method seeks a threshold $m_i$ which is the mean of averages $m_A$ and $m_B$ from two clusters divided by $m_i$: Leads (A) and pack ice (B)."

Ridler, T. W., and Calvard, S.: Picture thresholding using an iterative selection method. IEEE trans syst Man Cybern, 8(8),630-632, 1978.

"P6, L25-26: Is this difference or rather the larger number for MODIS really simply just due to mixed pixels? Later on the authors discuss frequently how much of the total area comes from small leads, which MODIS cannot detect at all. From reading the manuscript, I would rather expect it to be different as there should not be any leads in MODIS that Landsat-8 cannot detect, but surely as the authors also stated, the other way around. Could clouds be a factor here?"

Reply: Thanks for the question. Although the 1km resolution is the finest for MODIS thermal (and AVHRR), potential and subpixel lead can be detected by MODIS at this scale (Lindsay and Rothrock, 1995). The observed lead fractions are composite of thresholds and contrast in surface temperature of leads compared to the surrounding ice, i.e. temperature anomaly Δt. Mixed pixels at MODIS scale might be the main reason for the difference, but the threshold should also be considered. When high thresholds (2nd and 3rd Std) were applied, lead fraction extracted from MODIS drops quickly below that from TIRS (Table 4), this is consistent with result from Key et al. (1994).

About cloud contamination, as explained above, the Landsat-8 image area is mostly unobstructed by cloud. Difference in lead fraction caused by cloud is negligible here.

Page7, Line 24 ~ 26, we added:

"Although, the 1km resolution is the finest for MODIS thermal, the 1km wide lead category at MODIS scale should provide a reasonable guess of potential small leads or subpixel leads at MODIS scale (Lindsay and Rothrock, 1995)."

Page8, Line 33 ~ 34, we added:

"The obtained lead fractions are a composite of thresholds and contrast in surface temperature of leads compared to the surrounding ice, i.e., temperature anomaly Δt."

Page8, Line 38 ~ 40, we added:

"The difference in lead fractions from the two sensors mainly resulted from mixed pixels at MODIS scale, but the threshold should also be considered. When high thresholds (2nd and 3rd Std) are applied, the lead fraction extracted from MODIS drops quickly below that from TIRS (as shown in Table 4), and this is consistent with results from Key et al. (1994)."

Page 21, Line 3, we have Table 4:

**Table 4.** Threshold candidates for lead detection and corresponding lead fraction

|  |  | Fixed1 | Fixed2 | Fixed3 | 1st Std | 2nd Std | 3rd Std | Iterative |
|---|---|---|---|---|---|---|---|---|
| MODIS | Threshold (K) | 1 | 2 | 3 | 1.29 | 2.47 | 3.65 | 1.52 |
|  | Lead fraction (%) | 12.59 | 6.04 | 3.69 | 9.73 | 4.73 | 2.71 | 8.25 |
| TIRS | Threshold (K) | 1 | 2 | 3 | 1.90 | 3.52 | 5.14 | 2.49 |
|  | Lead fraction (%) | 14.85 | 8.65 | 6.62 | 8.93 | 5.69 | 2.82 | 7.53 |

Reference:

Key, J., Maslanik, J. A., and Ellefsen, E.: The effects of sensor field-of-view on the geometrical characteristics of sea ice leads and implications for large-area heat flux estimates. Remote sensing of environment, 48(3), 347-357, 1994.

Lindsay, R. W., and Rothrock, D. A.: Arctic sea ice leads from advanced very high resolution radiometer images.

Journal of Geophysical Research: Oceans, 100(C3), 4533-4544, 1995.

*"P6, L30: Is the choice of lead-width thresholds arbitrary or is there a reference for that from another study?"*

Reply: Thanks for raising this question. We do have some consideration when making these categories. According to Andreas and Cash (1999) model, turbulent heat flux over lead are stable for lead more than a few hundred meters wide. Therefore, for lead with width less than 1km, turbulent heat fluxes estimated from fetch-limited model and bulk formulae are expected to differ more than larger leads. While lead more than 5km or 10km wide is very large and rare in Arctic winter, but can be observed in marginal ice zone, indicating the beginning of summer ice retreat. The choice of 5km break point is partly due to the fact that lead at this scale can be easily observed by passive microwave sensors like AMSR-E/2, and coupled in climate models using bulk formulae. In comparison, we can see how much leads and turbulent heat flux from lead were missed if we used passive microwave data and bulk formulae alone.

Page 4, Line 38 ~ 39, we added:
"Using width samples crossed by transects, Lindsay and Rothrock (1995) found mean lead width between 2 and 3 km in the Arctic winter; larger means are found in peripheral seas."

*"P8, L16: Iterative thresholds are mentioned again but I think I still have not read an explanation yet."*
Reply: As explained above, a description was added; hope this time it is clear.

Page4, Line31~32, we added:
"Assuming an initial threshold using the mean temperature anomaly ($m_0$) of the whole image, the iterative method seeks a threshold $m_i$ which is the mean of averages $m_A$ and $m_B$ from two clusters divided by $m_i$: Leads (A) and pack ice (B)."

Ridler, T. W., and Calvard, S.: Picture thresholding using an iterative selection method. IEEE trans syst Man Cybern, 8(8),630-632, 1978.

*"P9, L19-20: Technically, MODIS cannot detect any leads in thermal infrared with a width below 1km? You compare numbers from below 1km width with numbers from exactly 1km. I think that should be highlighted better or rephrased."*
Reply: Thanks for raising this question. Yes, MODIS cannot directly detect leads with a width below 1km. But during our image processing, we found that the proportion of lead in a MODIS pixel will influence the finally classification of that pixel. In other words, subpixel leads might be detected at 1km using MODIS thermal images. In the revision, we include the 1km to the first category.

Page 9, Line 19~23, we added:
"In comparison with Landsat-8 TIRS and panchromatic images, we find that the lead map generated from the MODIS IST data neglects very small leads, but overestimates the width of other leads approximately 1 km wide. Overall, the 1 km wide lead category at MODIS scale should provide a reasonable guess of potential small or subpixel leads. The small leads retrieved using TIRS provide a valuable reference for the capacity of MODIS to detect narrow leads."

Page 10, Line 18~22, we modified the conclusion as:
"Within the studied area, the total length of leads is 10,150.3 km from TIRS, including 8502.2 km (83.76%) from small leads with width less than 1 km. This is in contrast to the total length of 2746.4 km from MODIS, where the narrow leads (1 km wide) only account for 1050.0 km (38.23%). The total length of leads is underestimated by

72.9% in the MODIS data. For the area of leads, small leads (width ≤ 1km) account for 34.54% of the total lead area from TIRS, but only 13.00% of the total lead area from MODIS."

**Technical Corrections:**
"P1, L11: I think that should be 'scales'"
Corrected.

"P1, L20: 'flux over leads'"
Corrected.

"P1, L23: 'exposed to the atmosphere'"
Corrected.

"P7, L6-7: I suggest to rephrase this sentence(s): Table 2 reveals that the total heat flux over leads calculated using TIRS IST is 6.59[: : :] over the total area of [: : :]km2. This is 42.33% larger [: : :]"
Corrected.

"P7, L14: Suggest to use 'difference' instead of 'increase'."
Corrected.

"P7, L18/19: 'leads' and 'widths'. To my understanding, there are probably quite some more cases of that throughout the manuscript. The authors should double-check that."
Corrected and checked.

"P7, L33-35: I find this last sentence hard to comprehend. Please rephrase."
Deleted.

"P7, L38: 'twice as' large?"
Corrected

"P8, L8: 'to extract lead signatures from the background'"
Corrected.

"P8, L24: Second Key reference is not capitalized."
Corrected.

**Reviewer 2:**

"A review on "Estimation of turbulent heat flux over leads using satellite thermal images"
    The focus of the paper is the estimation of turbulent sensible and latent heat fluxes over leads using high-resolution satellite thermal images. The heat transfer over leads play important role in the heat budget of the atmospheric and oceanic boundary layers and affects many processes in the Arctic climate system. However, there is a large uncertainty in the estimates of turbulent heat flux over leads due several reasons: i) the insufficient resolution of satellite images used in models, ii) sparseness of in situ observations and iii) uncertainties in parametrizations of turbulent heat transfer over inhomogeneous sea ice surface. The paper provides new estimates of such uncertainties using satellite images of various resolution and shows the necessity to use high-resolution images and also more adequate parametrizations. To some extent, the paper follows the line of the Marcq and Weiss

(2012) paper, but uses realistic surface and air temperatures, as well as wind speed for their case study and also using different satellite data. Therefore, the study adds to the current knowledge and provides revised estimates of the heat flux calculation uncertainties and thus is relevant and valuable.

However, the quality of the paper is low and has to be strongly improved. It concerns the choice and description of methods, the analysis of results and language. The paper cannot be accepted in its current state. I suggest major revision with resubmission."

**Major comments:**

"1.The two methods are used for the turbulent heat flux estimates: the traditional bulk formulae and the fetch-dependent model proposed by Andreas and Cash (1999).
(1) The bulk formulae and their application have to be described in more detail.
(2) First of all, it is potential temperature that has to be used in the formula for the sensible heat flux.
(3) Second, the heat transfer coefficients depend on height, surface roughness lengths for momentum and heat (z0m and z0t) and stability. Which height, which values for the roughness lengths and, finally, which universal stability functions are used?
(4) The authors say that they use the air temperature at 2m height, but wind speed at 10m height from the reanalysis data. Since these heights differ from each other, the bulk formulae cannot be used in their classical form.
(5) The authors need to describe in detail how they solve the bulk equations. Do they use z/L or the bulk Richardson number as a stratification parameter in the stability functions?"

Reply: Thanks for your questions and advices.

(1) As suggested, the bulk formulae are now modified as follow:

Page 5, Line13 ~ 17, we write:
"$F_s = \rho_a c_p C_{sh} u_r (T_s - T_r)$ (5)
$F_l = \rho_a L_v C_{le} u_r (Q_s - Q_r)$ (6)
where $\rho_a$ is the air density; $c_p$ is the specific heat at constant pressure; $L_v$ is the latent heat of vaporization; $u_r$, $T_r$, and $Q_r$ are wind speed, air temperature, and specific humidity at reference height $r = 2$ m, respectively; $T_s$ is surface temperature; and $Q_s$ is specific humidity close to the surface.."

Page 5, Line 24 ~ 25, we added:
"$C_{sh}$ and $C_{le}$ are transfer coefficients for sensible heat and latent heat, calculated using equations from Oberhuber (1988) and Goosse et al. (2000) (see Appendix B)."

(2) In our study, 2m air temperature from ERA-interim Reanalysis datasets was used as potential temperature $T_r$ in the formulae;
(3) A constant turbulent heat coefficient of $1.44 \times 10^{-3}$ from Nihashi and Ohshima (2001) was used in our previous experiment, which might not be appropriated for Arctic leads. As suggested, we have modified the experiments using equations (Appendix B) from Oberhuber (1988), Goose et al. (2000), and Marcq and Weiss (2012) to solve the coefficients for the bulk formulae. Most part of the parameterization originates from Large and Pond (1981, 1982):

Page 11, Line 2 ~ 18, we added Appendix B:
"Equations used for turbulent heat flux estimation using bulk formulae (Large and Pond, 1981, 1982; Oberhuber, 1988; Goosse et al, 2000; Marcq and Weiss, 2012):

$c_{sh} = 0.0327 \frac{k}{\ln(r/z_0)} \Phi_{sh} = c_{shN} \Phi_{sh}$ (B1)

$$c_{le} = 0.0346 \frac{k}{\ln(r/z_0)} \Phi_{le} = c_{leN} \Phi_{le} \tag{B2}$$

$$\Phi_{sh} = \frac{\sqrt{c_M/c_{MN}}}{1 - c_{shN} k^{-1} C_{MN}^{-1/2} \Psi_H} \tag{B3}$$

$$\Phi_{le} = \frac{\sqrt{c_M/c_{MN}}}{1 - c_{leN} k^{-1} C_{MN}^{-1/2} \Psi_L} \tag{B4}$$

$$\sqrt{\frac{c_M}{c_{MN}}} = \frac{1}{(1 - \sqrt{c_{MN}} k^{-1} \Psi_M)} \tag{B5}$$

$$c_{MN} = \frac{k^2}{\left(\ln\left(\frac{r}{z_0}\right)\right)^2} \tag{B6}$$

$$u_*^2 = c_M u_r^2 \tag{B7}$$

$$T_0 = T_r(1 + 2.2 \times 10^{-3} T_r q_r) \tag{B8}$$

Surface roughness lengths for momentum are given as:

$$z_0 = 0.032 \frac{u_*^2}{g} \tag{B9}$$

For unstable conditions:

$$\Psi_H(A) = \Psi_L(A) = 2 \ln\left(\frac{1 + A^2}{2}\right) \tag{B10}$$

$$\Psi_M(A) = 2 \ln\left(\frac{1 + A}{2}\right) + \ln\left(\frac{1 + A^2}{2}\right) - 2 \arctan A + \frac{\pi}{2} \tag{B11}$$

$$A = \left(1 - 16(r/L)\right)^{1/4} \tag{B12}$$

$$r/L = \frac{100r}{T_0 u_r^2} \left((T_s - T_r) + 2.2 \times 10^{-3} T_0^2 (q_s - q_r)\right) \tag{B13}"$$

(4) Since the wind speed and air temperature from ERA-interim are at different height, in our previous manuscript, a power law equation was used to calculate $u_r$ using 10m wind magnitude $u_{10}$ from ERA-interim:

$$\frac{u_r}{u_{10}} = \left(\frac{r}{10}\right)^a$$

where $a$ is the wind shear exponent. An empirical value of $a = 1/7$ was used. In our modified experiments, a log law equation based on principles of boundary layer flow, was used (Ray et al., 2006):

Page 5, line 26~29, we added:
Since the wind speed and air temperature from ERA-interim are at different heights, a wind profile equation was used (Ray et al., 2006):

$$\frac{u_{10}}{u_r} = \frac{\ln 10 - \ln Z_0}{\ln r - \ln Z_0} \tag{9}$$

where $u_{10}$ and $u_r$ are wind speed at 10m and 2m height, and $Z_0$ is the surface roughness length."

As a result, the mean wind speed at 2m height over leads, rises from ~5m/s (using power law) to ~7m/s (using log law). Estimated turbulent heat flux also increases in response.

(5) Assuming an initial value for friction velocity $u_*$, the equations (B1) ~ (B13) and Eq. (9) are solve iteratively. As shown in Eq. (B13), the equations use z/L as a stratification parameter, comparing to the bulk Richardson number used in Andreas and Cash (1999).

Page 5, Line38 ~ 40, we added:
    "Another mechanism is described in Esau (2007) for leads 1km~10km wide. Under weak wind condition (~2 m/s), convective overturning prevents cold breezes from penetrating into the lead area, reducing the average turbulent heat flux."

"3. As already mentioned, there is another model which takes into account the dependency of heat flux on fetch over leads, namely, the Alam and Curry (1997) model. This model has different physics and more processes are taken into account. It is not clear why the authors prefer the Andreas and Cash model and do not consider the Alam and Curry model. This has to be explained."
Reply: Thanks for pointing out the Alam and Curry (1997) model. In that model, turbulent heat flux across single lead is integrated from fetch 0 to fetch X, along the wind direction. Indeed, it is more theoretical and takes more factors into account. But in the large scale application, it is hardly applicable due to lack of high resolution meteorological data like 2m air temperature. Besides, the model assumes universal water surface within leads (with a complicated surface roughness model for open ocean), which is different from our case where narrow open water or thin ice dominate. Actually, we tried to apply the Alam and Curry (1997) model in the remote sensing setting, but failed due to lack of sea surface information (e.g. phase speed, wave age etc.). Thus, only the fetch-limited model of Andreas and Cash (1999) was selected in our experiment.

Page 6, Line 2 ~ 8, we added the explanations as follows:
"However, the assumption of universal water surface in leads and the application of sea surface roughness model (Andreas and Murphy, 1986; Alam and Curry, 1997) are not applicable in our case, where open water and thin ice dominate. Since the signal of TIBL is absent in the coarse grid of 2 m air temperature from the ERA reanalysis dataset, the data might not be appropriate to demonstrate the Alam and Curry (1997) model, which relies on accurate measurement of meteorological parameters. Whereas the Andreas and Cash (1999) model is more sensitive to lead width than atmospheric conditions (Marcq and Weiss, 2012). Therefore, only the Andreas and Cash (1999) model was used in our experiment."

"4. One of the results of the study is that the fetch-dependent model produces larger fluxes than the bulk formulae. However, the transfer coefficients in the bulk formulae depend strongly on the roughness length for momentum and heat (z0m and z0t) and therefore, the obtained result is only valid for specific values of z0m and z0t, which are not given in the paper (!). Using other values for z0m and z0t can produce completely different results.
    The Andreas and Cash model, as it is described in the paper, does not show an explicit dependency on the roughness length. However, implicitly, roughness is present in their model and the authors need to describe how the roughness length is present in the model of Andreas and Cash. What are the values for the roughness length in the Andreas and Cash model and how do they compare with the ones used in the bulk formulae?
    Note, that the Andreas and Cash model is a reformulation of the earlier Andreas and Murphy model. The latter is formulated in such a way that it is compatible with bulk formulae. Namely, they are suggesting to use a fetch-dependent "lead-averaged" neutral heat transfer coefficient. In other words, the Andreas and Murphy formulation would allow a more reasonable comparison with the standard bulk approach."
Reply: Thanks for the question. For open water leads, Andreas and Murphy (1986) parameterize turbulent transfer coefficient at 10m as $10^3 C_{N10} = 1.0 + 0.8\,EXP\,(0.05(X/L))$, where $X$ is the fetch or lead width. However, the reference

height of 10m might not be suitable for study narrow leads in pack ice, where the TIBL is generally shallower. On the base of Andreas and Murphy (1986), Andreas and Cash (1999) used bulk Richardson number $Ri_b$ to calculate Obukhov Length $L$ i.e. Eq. (17) under the assumption that the ratio between roughness lengths for momentum and heat, i.e. $Z_0 / Z_T$, is about $e^2$. In our case, this simplification result in a higher $|L|$ than that from Oberhuber (1988) and Goosse et al. (2000). If the Obukhov length from Eq. (B8) and (B13) were used to calculate the coefficient $C_*$ in Andreas and Cash (1999) model, and estimated turbulent heat flux will be smaller (Table 3), but still 15.53% larger than that from bulk formulae.

Page 9 line 25~31, we added:

"In both the Andreas and Murphy (1986) and Andreas and Cash (1999) models, for reference height r < 10 m, the ratio between roughness lengths for momentum and heat, $Z_0/Z_T$, is assumed to be ~$e^2$ to calculate Obukhov length $L$ using Richardson number $Ri_b$ (see Eq. (17)). The calculated Obukhov length L has absolute values about 67% higher than those using Eq. (B8) and (B13) from the bulk formulae (Oberhuber, 1988; Goosse et al., 2000). If Eq. (B8 ) and (B13) were used to solve Obukhov length and coefficient $C_*$ in the Andreas and Cash (1999) model, estimated turbulent heat flux will be smaller (Table 3), but still 15.53% larger than that from the bulk formulae, with an even larger part of the difference from the small lead category (42.48%, compared to 32.96% in Section 4.3.2)."

Page 6, Line 27, we have Eq. (17)

$$L^{-1}=8.0 * \left(\frac{0.65}{r} + 0.079 - 0.0043r\right)*Ri_b \qquad (17)$$

Page 21, we updated Table 3:

**Table 1.** Same as Table 3 in the manuscript, estimated turbulent heat flux (W) for Landsat-8 TIRS using bulk formulae, the Andreas and Cash (1999) model, and modified Andreas and Cash model using Obukhov length from Eq. (B8) and (B13).

| Lead category | Bulk formulae | | Andreas and Cash & L | | Andreas and Cash (1999) | |
|---|---|---|---|---|---|---|
| | Heat flux | Contribution | Heat flux | Contribution | Heat flux | Contribution |
| <1km | 2.16E+11 | 25.75% | 2.72E+11 | 27.99% | 3.06E+11 | 27.50% |
| 1km~5km | 3.37E+11 | 40.09% | 3.86E+11 | 39.75% | 4.43E+11 | 39.81% |
| >5km | 2.87E+11 | 34.17% | 3.13E+11 | 32.25% | 3.63E+11 | 32.68% |
| Total | 8.40E+11 | | 9.71E+11 | | 1.11E+12 | |

"5. Describe better the case study. Which date is it, what are the synoptic conditions over the study area? Was it a clear-sky case or clouds were present? Does it represent typical conditions in the Arctic? The presented surface and air temperature distributions suggest that this is either autumn or late spring. But one would expect that the effect of leads is strongest in winter."

Reply: Thanks for the question. The Landsat-8 and MODIS images used in our study were acquired on April 26, 2016. The study area is mostly unobstructed by cloud in this scene. According to the ERA-interim datasets, the study area is dominated by polar easterlies, with 10m wind speed range from 4.8 to 9.5 m/s. Air temperatures at 2m from the reanalysis data range from 257.3 ~ 263.8K. Temperature difference between surface and 2m air is about 5K. The Landsat-8 imagery is sparse in the Arctic ocean, especially in the winter polar night. These three successive scenes of thermal images can also provide valuable details of spatial distribution of spring leads.

Page 3, Line 6, we added: "acquired on April 26, 2016"

Page 20, we updated Table 1:

**Table 2.** Same as Table 1 in the manuscript, satellite images and other data used in this study.

| Resource | Parameters | Spatial-resolution | Time | Notes |
|---|---|---|---|---|
| Landsat-8 TIRS | Band5 | 30m | 21:27 | Near-infrared |
| | Band8 | 15m | 21:27 | Panchromatic |
| | Band10 | 30m | 21:27 | 10.60μm-11.19μm |
| | Band11 | 30m | 21:27 | 11.50μm-12.51μm |
| Terra MODIS | Band31 | 1000m | 20:55 | 10.78μm–11.28μm |
| | Band32 | 1000m | 20:55 | 11.77μm–12.27μm |
| ERA-interim Reanalysis | 10m wind | 0.125°(~10km) | 21:00 | 4.8~9.5m/s |
| | 2m air temperature | 0.125°(~10km) | 21:00 | 259.3~265.6K |
| | 2m dew point temperature | 0.125°(~10km) | 21:00 | 257.3~263.8K |

**Minor comments:**

–Page 1, lines 26-27, rephrase "The rate of turbulent heat transferred", simply "Turbulent heat flux"

Corrected

–Page 1, line 36. "More intensive network" needs clarification. Also, "stronger influence of leads" - influence on what?

Corrected, "networks of more intensive lead with stronger local influence are expected"

– Page 2, line 14 - "heat flux transfer rate" - the efficiency of heat transfer?

Corrected, "Assuming higher heat transfer over narrow leads than wider leads"

– Page 2 line 16 – remove "More often than not"

Corrected

– Page 2, lines 20-25, explain better what is meant by "Fetch limited models" and how they are using the fetch-dependence of the internal boundary layer height. Otherwise, the logic is disrupted.

A review of models developed for estimation of internal boundary layer height and turbulent heat flux was added on page 2 Line 27 ~ 40.

– At least in the introduction the authors should cite the study by Tetzlaff et al. (2015) where the most recent observations of heat fluxes and the internal boundary layer height over leads are presented: Tetzlaff, A. , Lüpkes, C. and Hartmann, J. (2015), Aircraft based observations of atmospheric boundary layer modification over Arctic leads. Q.J.R. Meteorol. Soc., 141: 2839-2856. doi:10.1002/qj.2568

Page2, Line 25 ~ 26, we added:

"Convective plumes formed above leads may further complicate the process within the TIBL (Tetzlaff et al., 2015)."

– Page 3, line 10. The actual grid of the ERA-Interim reanalysis has horizontal spacing 0.75o and not 0.125o. The original ERA Interim data is interpolated on the 0.125o grid which does not increase the resolution.

Corrected

Page3, Line 9, we write: "This dataset provides global coverage with a temporal resolution of 3 hours, 0.125° grid data is available for download (~10 km in study area, interpolated from original 0.75° grid)."

– Page 3, line 39. albedo anomaly

Corrected

– Page 4, Lines 8-9 rephrase "varying air condition"

Corrected, "air temperature variation"

– Page 4, Line 21. "limited used of lead width" - what does it mean??
Deleted

– Page 4, Line 29 "rate of turbulent heat change" - what does it mean?? Rephrase!
Corrected, "turbulent heat exchange"

– Page 4, Eq. 4 – use CH instead of Cs for the heat transfer coefficient
Corrected

– Page 4, Line 36 Ce is the bulk transfer coefficient for water vapor
Corrected

– Page 4, Line 37 "at the surface", this is wrong. These values are at heights z0t and z0q, which are the roughness lengths for heat and moisture fluxes.
Corrected.

– Page 4, Do the authors use the saturation specific humidity over ice or over water? Do they distinguish between the open water and thin ice in this respect?
Yes. Saturated specific humidity was used over lead surface for both open water and thin ice, but different sets of parameter were used to calculate saturated vapor pressure at the surface of open water and thin ice.
Page 5, Line 20~23, we write:
"$e_{s0}$ represents the saturated vapor pressure at surface temperature $T_s$:

$$e_{s0} = e_0 10^{\frac{at}{b+t}} \tag{8}$$

with $e_0$ represent saturate vapor pressure at 0 °C, approximately 6.11 hPa, $t$ is temperature in Celsius, and $a$ and $b$ are coefficients (for water surface, $a = 7.5$, $b = 237.3$ K; for ice, $a = 9.5$, $b = 265.5$ K)."

– Page 5, Line 11. This is shown in figure 4, not in figure 3.
Corrected

– Page 5, Line 35, Not "in spite", but "apart from a dependency on the width of a lead"
Corrected

– Page 5, line 36. Which simulation? What was used for this simulation - Bulk formulae, or the Andreas and Cash model? For the bulk model, such result is obvious and does not need to be shown. Andreas and Cash write that their model is, on the contrary, not very sensitive to wind speed.
The simulation, as well as Fig. 3 and 4, is based on Andreas and Cash (1999) parameterization. Although, they claim that their model "depends only weakly on surface level wind speed". Our test shows that, for the narrowest lead from TIRS (X=30m), turbulent heat flux, especially sensible heat, rises quickly with larger Δt and stronger wind.

– Page 6, Section name "Results" and not "Result".
Corrected

– Page 6, line 35. How is the length of leads calculated?

Page 4, Line 43 ~ page 5, Line 2 we added:

Since we assign lead width to every pixel across the lead, the length $L_i$ for lead width $X_i$ can be calculated as follow:

$$L_i = \frac{a_0 N_i}{X_i} \tag{4}$$

where $a_0$ is the pixel size, for TIRS, the value is 30 m, for MODIS, 1km; and $N_i$ is pixel number for width $X_i = a_0 i, (i = 1, 2, 3\ldots)$.

– Page 6, line 35. The Authors say that the MODIS resolution is 1000m. However, they introduce a class of small (which they call narrow in other places) leads with width less or equal 1km. Somehow, they found 13% of such leads in the MODIS image. They need to comment if 1km wide leads are resolved with 1km resolution.

Reply: The detectability of leads are composite of thresholds and contrast in surface temperature of leads compared to the surrounding ice, i.e. temperature anomaly Δt. Although the 1km resolution is the finest for MODIS thermal (and AVHRR), potential and subpixel lead can be detected at this scale (Lindsay and Rothrock, 1995). In the revised version, we now include 1km in the first category, and statistics in Table 2 are updated.

Page7, Line 24 ~ 26, we added:

"Although, the 1km resolution is the finest for MODIS thermal, the 1km-wide lead category should provide a reasonable guess of potential small leads or subpixel leads at MODIS scale (Lindsay and Rothrock, 1995)."

Page 9, Line19 ~ 23, we added:

"In comparison with Landsat-8 TIRS and panchromatic images, we find that the lead map generated from the MODIS IST data neglects very small leads, but overestimates the width of other leads approximately 1 km wide. Overall, the 1 km wide lead category at MODIS scale should provide a reasonable guess of potential small or subpixel leads. The small leads retrieved using TIRS provide a valuable reference for the capacity of MODIS to detect narrow leads."

Lindsay, R. W., and Rothrock, D. A.: Arctic sea ice leads from advanced very high resolution radiometer images. Journal of Geophysical Research: Oceans, 100(C3), 4533-4544, 1995.

– Page 7, line 4. The direction of fluxes from the ocean to the atmosphere is not consistent with the Eq. (4) and (5). The authors should modify the Equations to get the right sign and direction of fluxes.

Corrected.

– Page 7 line 12. In which range of width is the fetch-limited model valid? Why was it applied to Landsat only? There wide leads in Landsat as well. Write that Landsat better resolves leads, so that's why it was applied to it.

Reply: Andreas and Cash (1999) fit $C_*$ for -h/L range from 0.05 to 20. Although high wind (~7m/s) and low temperature difference (~5K) in our case leading to large |L| up to 40m, the $C_*$ fitting still holds for lead width X > 15m from TIRS. Since the turbulent heat flux saturate for lead width great than 1km, as depicted in Fig. 4, we think there is no need to apply the model with coarse leads from MODIS.

**Marked-up manuscript:**

[revised manuscript text omitted]

---

## Author Response (AR2)

Dear Editor:

Many thanks for the review of our manuscript entitled "Estimation of turbulent heat flux over leads using satellite thermal images" (tc-2018-262)

We have rechecked the manuscript as suggested. Please find attached a list of changes made in the manuscript, and a revised version of our manuscript, in which all relevant changes are marked in red. We hope that the revised version of the manuscript is now acceptable for publication in your journal.

Looking forward to hearing from you soon.

Best Regards,

Meng Qu, on behalf of all the authors

**List of changes:**

Page 4, Line 36, we added, "(see section 5.1), and iterative threshold method was adopted."

Page 4, Line 41, we added, "$x$"

Page 4, Line 44, we corrected the Eq. (4) as:

$$L_i = \frac{a_0^2 N_i}{X_i} = \frac{a_0 N_i}{i} \tag{4}$$

Page 5, Line 26, we added, "to calculate wind speed at 2 m height"

Page 6, Line 17, We added, "$g$ is acceleration due to gravity;"

Page 7, Line 26, we chose "relating to" instead of "using"

Page 7, Line 33, we used "no more than" instead of "less than"

Page 8, Line 6 and Line 11, we used "Andreas and Cash (1999) model" instead of "Fetch-limited model"

Page 10, Line 7, we added, "lead width of"

Page 12, Line 14, we added, "Air pressure: $p$ = 101 kPa"

Page 12, Line 22, we added, "Von Karman constant: $k$ = 0.4"

Page 15, Line 8, we added, "$x$ is the real lead width."

Page 15, Line 17, we used "Spatial distribution of heat flux derived from" instead of "Heat flux from"

Page 20, Table 2, total lead length from MODIS was corrected, other statistics are not affected.

Page 20 ~ 21, we modified the arrangements of statistics in Table 2 and Table 3.

Variables and references were checked for consistency and correctness, related figures were also updated.

[revised manuscript text omitted]

---

## Author Response (AR3)

Dear editor:

Many thanks for your acceptation of our manuscript entitled "Estimation of turbulent heat flux over leads using satellite thermal images" (tc-2018-262)

We have uploaded all files required. A few more minor changes were made in the updated version of our manuscript, and a short list is provided as below. Please find attached an updated version of our manuscript, in which all relevant changes are marked in red.

Page 1, Line 5, we added another affiliation, "University Corporation for Polar Research, Beijing, China", for coauthor Xiaoping Pang and Xi Zhao
Page 3, Line 6, the year of image acquisition was corrected to be "2015"
Page 3, Line 36, for consistency, we used "$T_s$" in Eq. (1) instead of "IST"
Page 11, Line 1, in acknowledgement, we added another National Key Research and Development Program of China, "No. 2018YFC1407100"

Looking forward to hearing from you soon.

Best Regards,

Meng Qu, on behalf of all the authors

[revised manuscript text omitted]